# ALIGN, DON'T DIVIDE: REVISITING THE LoRA ARCHITECTURE IN MULTI-TASK LEARNING

## ABSTRACT

Parameter-Efficient Fine-Tuning (PEFT) is essential for adapting Large Language Models (LLMs). In practice, LLMs are often required to handle a diverse set of tasks from multiple domains, a scenario naturally addressed by multi-task learning (MTL). Within this MTL context, a prevailing trend involves LoRA variants with multiple adapters or heads, which advocate for structural diversity to capture task-specific knowledge. Our findings present a direct challenge to this paradigm. We first show that a simplified multi-head architecture with high inter-head similarity substantially outperforms complex multi-adapter and multi-head systems. This leads us to question the multi-component paradigm itself, and we further demonstrate that a standard single-adapter LoRA, with a sufficiently increased rank, also achieves highly competitive performance. These results lead us to a new hypothesis: learning task-shared representations provides a highly effective and promising path towards multi-task learning, offering a powerful alternative to the architectural isolation of task-specific features. To validate this, we propose Align-LoRA, which incorporates an explicit loss to align task representations within the shared adapter space. Theoretical analysis and experiments confirm that Align-LoRA significantly surpasses baselines, establishing a simpler yet more effective paradigm for adapting LLMs to multiple tasks. The code is available anonymously.

## 1 INTRODUCTION

In recent years, large language models (LLMs) have demonstrated unprecedented performance across a wide range of natural language processing (NLP) tasks (Brown, 2020; Zhao et al., 2023; Chang et al., 2024b). Despite their strong generalization abilities, LLMs often require further adaptation to align with domain-specific requirements or to incorporate updated knowledge (Agiza et al., 2024; Xin et al., 2024). Supervised fine-tuning (SFT) plays a critical role in this process, but full parameter fine-tuning (FFT), which updates all model parameters, poses significant challenges in terms of computational and memory costs (Mao et al., 2025).

To address these demands, parameter-efficient fine-tuning (PEFT) methods have been proposed to adapt LLMs by updating only a small subset of parameters (Han et al., 2024; Chang et al., 2024a). Among these, Low-Rank Adaptation (LoRA) (Hu et al., 2021) has become a widely adopted approach. It approximates the full-rank weight update matrix by decomposing it into two low-rank matrices: a down-projection matrix $\mathbf{A}$ and an up-projection matrix $\mathbf{B}$. In practice, adapting LLMs often involves data from multiple domains or tasks, naturally aligning with the multi-task learning (MTL) paradigm.

Consequently, this has motivated the development of LoRA variants specifically designed for MTL. An early approach is the Multi-Adapter architecture, which employs multiple, distinct pairs of down-projection ($\mathbf{A}$) and up-projection ($\mathbf{B}$) matrices for different tasks (Wang et al., 2023). To improve parameter efficiency, the Multi-Head architecture was introduced, typically sharing a single $\mathbf{A}$ matrix while maintaining multiple task-specific head matrices ($\mathbf{B}$) (Tian et al., 2024). Furthermore, many of these multi-component architectures employ a routing mechanism, inspired by the Mixture-of-Experts (MoE) framework, to dynamically select or weigh the outputs of different adapters for a given input. Recent prevalent methods like R-LoRA (Liu et al., 2025) further refine this by explicitly encouraging diversity among heads to mitigate redundancy. **Despite architectural differences,**

**these methods are all built on a common premise: that effective multi-task adaptation requires architectural isolation of task-specific knowledge.**

However, this pursuit of architectural isolation introduces a significant practical drawback. The specialized components cannot be merged into the backbone model, resulting in non-negligible inference latency from their processing in every forward pass. Motivated by this trade-off, our work begins with an empirical re-evaluation of this premise, seeking a multi-task adaptation method that eliminates such a latency penalty. We first reveal a paradoxical finding: by simplifying a complex multi-head architecture into a model we term M-LoRA (which removes the dynamic router), we observe that its performance surpasses its more complex counterparts. This occurs despite the simplified model exhibiting higher inter-head similarity, a result that directly challenges the prevailing assumption that component diversity is beneficial. This outcome led us to a more fundamental question: **Is the multi-component structure truly necessary for effective multi-task adaptation?**

In pursuit of an answer, we discovered that merely increasing the rank of a standard, single-adapter LoRA is sufficient to match or even outperform these intricate multi-component variants. Collectively, the findings that a simplified multi-head model excels and that a high-rank single-head model is equally or more effective point to a new and unexplored hypothesis: **learning task-shared representations provides a highly effective and promising path towards multi-task learning, offering a powerful alternative to the architectural isolation of task-specific features.** To directly validate this hypothesis and operationalize this principle, we propose Align-LoRA. This method enhances a standard LoRA by augmenting its training objective with a component based on the Kullback-Leibler (KL) Divergence (Kullback & Leibler, 1951), which encourages the alignment of task representations in the shared low-rank space without adding parameters or inference overhead.

Our key contributions are fourfold:

- We demonstrate that a simplified multi-head LoRA (**M-LoRA**) with high head similarity outperforms complex variants, challenging the prevailing assumption that architectural isolation of task-specific knowledge is necessary.

- We show that simply **increasing the rank** of a standard LoRA can match the performance of multi-component architectures, questioning their fundamental necessity for multi-task learning.

- We propose a new hypothesis: learning task-shared representations provides a highly effective and promising path towards multi-task learning, offering a powerful alternative to the architectural isolation of task-specific features.

- We introduce **Align-LoRA**, a novel method that validates our hypothesis by explicitly aligning representations, achieving superior performance and setting a new direction for multi-task PEFT.

## 2 RELATED WORKS

### 2.1 LOW-RANK ADAPTATION (LoRA)

Current LLMs typically adopt a decoder-only architecture, consisting of stacked transformer blocks (Zhao et al., 2023). Each block contains two core components with residual connections: a multi-head self-attention (MHA) layer and a feed-forward network (FFN) (Vaswani, 2017). Both layers rely on dense learnable weight matrices $\mathbf{W}$ for feature transformation.

To efficiently adapt LLMs under resource constraints, LoRA (Hu et al., 2021) offers an effective solution. It is inspired by the hypothesis that the intrinsic dimensionality of weight updates in LLMs is low. LoRA approximates the weight update $\Delta\mathbf{W}$ using two low-rank matrices $\mathbf{A} \in \mathbb{R}^{r \times n}$ and $\mathbf{B} \in \mathbb{R}^{m \times r}$, where $\mathbf{W} \in \mathbb{R}^{m \times n}$ is the original weight matrix. The rank $r$ is chosen to be significantly smaller than $\min(m, n)$, reducing the number of trainable parameters from $\mathcal{O}(mn)$ to $\mathcal{O}(r(m+n))$. The forward pass is modified as follows:

$$h = (\mathbf{W} + \Delta\mathbf{W})x = \mathbf{W}x + \mathbf{BA}x, \tag{1}$$

where $\Delta\mathbf{W} = \mathbf{BA}$ denotes the low-rank update. A key advantage of LoRA is that after training, the low-rank update $\Delta\mathbf{W}$ can be merged back into the original weights $\mathbf{W}$, introducing zero inference overhead.

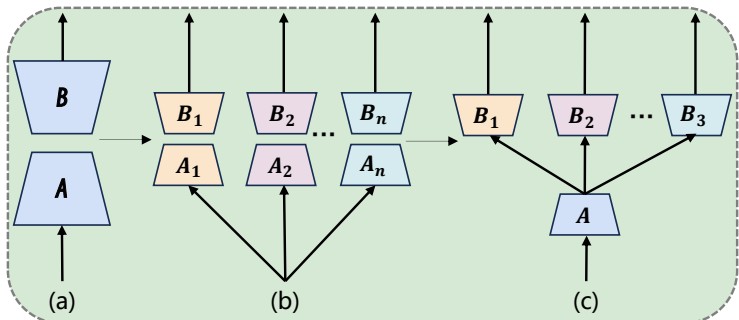

Figure 1: A comparison of LoRA architectural paradigms: (a) Vanilla LoRA; (b) the Multi-Adapter framework; and (c) the Multi-Head framework. The categorization of the multi-component architectures is adapted from R-LoRA (Liu et al., 2025). A common feature of these multi-component designs is the inclusion of a dynamic routing mechanism.

Several works have built upon the original LoRA framework. AdaLoRA (Zhang et al., 2023) dynamically allocates the rank budget, while DoRA (Liu et al., 2024b) decomposes weight updates into magnitude and direction. Other methods like PiSSA (Meng et al., 2025) and NLoRA (Guo et al., 2025) have focused on improving performance through better initialization and decomposition strategies, highlighting the ongoing effort to enhance LoRA's effectiveness.

## 2.2 MULTI-COMPONENT LoRA

To adapt LoRA for multi-task learning (MTL), a natural extension is to employ multiple trainable components. Early works proposed the **Multi-Adapter** architecture, which utilizes multiple independent LoRA adapters (i.e., distinct $\mathbf{B}_i\mathbf{A}_i$ pairs) for different tasks. Notable examples of this approach include Multi-LoRA (Wang et al., 2023), MixLoRA (Li et al., 2024), LoRAMoE (Dou et al., 2023), MoELoRA (Liu et al., 2024a), and LoRAHub (Huang et al., 2023).

The **Multi-Head** architecture was developed to improve parameter efficiency, driven by the key insight that LoRA's matrices have distinct roles. It was observed that down-projection matrices (**A**) capture redundant, **task-general knowledge**, while up-projection matrices (**B**) learn diverse, **task-specific features**. Consequently, the Multi-Head design, exemplified by methods like HydraLoRA (Tian et al., 2024), MALoRA (Wang et al., 2024), MTLLoRA (Agiza et al., 2024), and R-LoRA (Liu et al., 2025), employs a single shared **A** matrix with multiple distinct $\mathbf{B}_i$ heads. To further enhance task specialization within this paradigm, R-LoRA introduced a randomization technique to also reduce similarity among the head matrices. Figure 1 illustrates the architectural differences between three key paradigms: the original LoRA, the multi-adapter architecture, and the multi-head architecture.

The **Multi-Head** architecture, exemplified by methods like HydraLoRA (Tian et al., 2024) and R-LoRA (Liu et al., 2025), uses a shared down-projection matrix **A** and multiple head matrices $\mathbf{B}_i$. The aggregated weight update in this structure is a dynamically weighted sum of each head's output:

$$\Delta\mathbf{W} = \sum_{i=1}^{N} \omega_i(\mathbf{x}) \cdot \mathbf{B}_i\mathbf{A}. \tag{2}$$

Drawing inspiration from the Mixture-of-Experts (MoE) framework, this dynamic routing mechanism employs a learnable routing matrix $\mathbf{W}_r$ and a gating function, such as softmax or Top-K, to assign weights to each "expert" adapter based on the input $\mathbf{x}$. The widely used softmax-based router is formulated as:

$$\boldsymbol{\omega}(\mathbf{x}) = \text{Softmax}(\mathbf{W}_r\mathbf{x}). \tag{3}$$

However, this complexity introduces a critical trade-off. A significant drawback of input-dependent routing is that the aggregated update $\Delta\mathbf{W}$ can no longer be pre-computed. Consequently, the adapter weights **cannot be merged** into the frozen backbone model post-training. This results in **non-negligible inference latency**, as the router and multiple heads must be processed for each forward pass, sacrificing one of LoRA's most significant practical advantages.

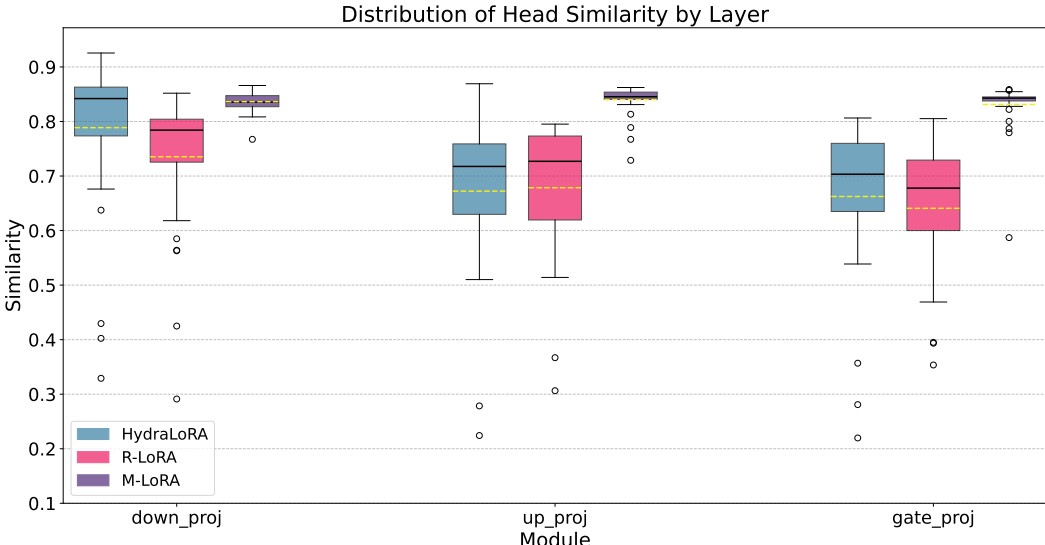

Figure 2: Distribution of inter-head cosine similarity across different model modules. Within each module, the box plots from left to right correspond to HydraLoRA, R-LoRA, and M-LoRA, respectively. The solid line within each box indicates the median similarity, while the dashed line represents the mean.

## 3 OBSERVATIONS

In this section, we critically examine the prevailing assumption that component diversity are essential for effective multi-task adaptation with LoRA. By questioning the fundamental necessity of the prevalent multi-head paradigm, our investigation leads to a new hypothesis centered on the pivotal role of shared knowledge.

### 3.1 M-LoRA: A SIMPLIFIED VARIANT

Prevalent methods like R-LoRA (Liu et al., 2025) are built on the premise that encouraging diversity among adapter heads is crucial for capturing distinct, task-specific knowledge. To directly test the hypothesis on head diversity, we propose **M-LoRA** (Base Multi-Head LoRA), a minimal ablation variant of R-LoRA. While preserving R-LoRA's core designs, such as multi-head randomization for initialization and input differentiation via Dropout, M-LoRA's sole architectural change is the removal of the dynamic routing module. Instead, it aggregates the outputs of its head matrices by simple summation, allowing us to directly study the effect of eliminating explicit, input-dependent diversification. The framework of M-LoRA is provided in the Appendix B.

### 3.2 THE PARADOX OF DIVERSITY: LESS IS MORE

We fine-tune the Qwen2.5-3B (Qwen Team, 2024) model using HydraLoRA, R-LoRA, and M-LoRA on a benchmark comprising five distinct tasks: QNLI (Wang, 2018), PiQA (Bisk et al., 2020), Winogrande (Sakaguchi et al., 2021), ARC (easy & challenge) (Clark et al., 2018), and GSM8K (Cobbe et al., 2021). To quantify inter-head similarity, we compute a matrix of pairwise cosine similarities between all flattened head vectors ($\mathbf{B}_i$). The final metric is the mean of this matrix's off-diagonal values. All Experimental details in this work, including implementation specifics, hyperparameter settings, dataset descriptions, baseline configurations, and other relevant information, are documented in the Appendix G.

Our findings reveal a paradox regarding the role of head diversity in multi-task adaptation. Figure 2, which plots the inter-head cosine similarity, shows that R-LoRA successfully achieves its design goal of maximizing diversity, exhibiting the lowest similarity. In stark contrast, M-LoRA, which lacks any diversity-enforcing mechanism, displays the opposite effect, yielding a high degree

| Schemes | QNLI | PiQA | Winogrande | ARC | GSM8K | Avg | %Para |
|---|---|---|---|---|---|---|---|
| HydraLoRA | 81.91 | 84.21 | 70.92 | 87.21 | 45.95 | 74.04 | 0.45 |
| w/o Router | 81.33 | 83.51 | 70.14 | 86.46 | 45.35 | 73.58 | 0.41 |
| R-LoRA | 82.03 | 85.55 | 71.84 | 87.69 | 46.25 | 74.67 | 0.45 |
| M-LoRA | **82.52** | **86.76** | **72.95** | **88.15** | **46.85** | **75.45** | **0.41** |

Table 1: Comparative study of several multi-head LoRA variants across five tasks.

of head redundancy with similarity medians consistently exceeding 0.85. Paradoxically, as demonstrated in Table 1, this high-redundancy model achieves superior multi-task performance. Despite its architectural simplicity, M-LoRA consistently and significantly outperforms the more complex HydraLoRA and R-LoRA. This outcome presents a fundamental contradiction to the philosophy of prior work: the architectural configuration that seemingly violates the principle of head diversity actually enhances multi-task generalization.

### 3.3 TASK-SHARED VS. TASK-SPECIFIC LEARNING

This section seeks to explain the surprising effectiveness of M-LoRA. Its strong multi-task generalization, achieved despite high inter-head similarity, challenges conventional assumptions and offers a new perspective on the core principles of multi-task adaptation in LoRA. Improving multi-task learning (MTL) has largely followed two distinct paths: isolating **task-specific** knowledge to mitigate interference, or enhancing **task-shared** knowledge to improve generalization. To date, the predominant focus has been on the former. Recent multi-task LoRA methods, such as LoRA MoE (Dou et al., 2023) and R-LoRA (Liu et al., 2025), have predominantly focused on isolating task-specific knowledge. **In contrast, the alternative path of actively enhancing task-shared knowledge within the LoRA framework has remained unexplored.**

**M-LoRA** challenges this specialization-focused paradigm. We hypothesize that the high similarity is not a sign of failed specialization, but a feature derived from the architecture's implicit regularization. The key mechanism is the interplay between removing the router and retaining the multi-head dropout. In models like R-LoRA, the heads are treated as competing "specialists," and the dynamic router attempts to select the single best expert, which can often lead to redundancy or load imbalance. In contrast, by replacing the dynamic router with simple averaging, M-LoRA compels the multiple **B** heads to form a collaborative ensemble. As illustrated in Figure 4, M-LoRA achieves this by retaining the multi-head dropout mechanism from R-LoRA (Liu et al., 2025). The dropout forces each head to learn from a slightly different input perspective. By forcing all heads to contribute (via summation), they are compelled to converge on a robust, task-general representation that works well from all perspectives.

To validate this mechanism, we performed an ablation on a non-dropout multi-head variant, HydraLoRA. As shown in Table 1, removing the router from HydraLoRA ('w/o Router') causes its average performance to drop, while M-LoRA's collaborative ensemble achieves the highest performance among tested variants. This strongly confirms that the multi-head dropout is the critical factor that, when combined with router removal, transforms the heads from isolated "specialists" into effective "collaborators" and significantly enhances task-general learning.

The success of M-LoRA suggests a revised viewpoint: **learning task-shared representations provides a highly effective and promising path towards multi-task learning, offering a powerful alternative to the architectural isolation of task-specific features.**

## 4 INCREASING RANK: A UNIFIED ADAPTER

Given M-LoRA's strong performance with highly redundant heads, which largely learn shared knowledge, this section explores a critical question: Is the multi-head architecture itself truly necessary for multi-task generalization, or does it merely serve as a means to increase total trainable parameters rather than offering genuine benefits?

To test this, we design a straightforward yet powerful experiment. We abandon the multi-component structure entirely and instead use a standard, single-adapter LoRA. We reallocate the entire parame-

Table 2: Comparison of different training schemes on LLaMA2. All variants use a LoRA rank of 8, except LoRA$^\dagger$(rank=30, adjusted to match multi-head variants' trainable parameter count). * indicates results from Tian et al. (2024).

| Metrics | Base | LoRA | LoRAHub* | LoRA MoE* | HydraLoRA | R-LoRA | LoRA$^\dagger$ | M-LoRA |
|---|---|---|---|---|---|---|---|---|
| 7B | 31.61 | 37.05 | 39.70 | 40.30 | 41.46 | 42.24 | 42.21 | **42.83** |
| 13B | 38.42 | 40.73 | 41.90 | 43.70 | 44.31 | 44.96 | 45.02 | **46.16** |
| % Param | - | 0.06 | 1.24 | 2.98 | 0.34 | 0.34 | 0.34 | 0.32 |

Table 3: Comparison of different training schemes on Qwen2.5. The superscript in "LoRA" (e.g., $^4$, $^8$, etc.) indicates the rank value used for each variant.

| Metrics | Base | LoRA$^4$ | LoRA$^8$ | LoRA$^9$ | LoRA$^{10}$ | HydraLoRA | R-LoRA | M-LoRA |
|---|---|---|---|---|---|---|---|---|
| 7B | 39.82 | 43.21 | 46.66 | 48.18 | 49.51 | 49.12 | 49.51 | **49.74** |
| 14B | 45.33 | 48.18 | 51.82 | 52.74 | **54.23** | 53.76 | 54.08 | 54.18 |
| Rank | - | 4 | 8 | 9 | 10 | 4 | 4 | 4 |
| % Param | - | 0.10 | 0.20 | 0.22 | 0.25 | 0.25 | 0.25 | 0.22 |

ter budget of the complex variants into this single adapter by simply increasing its rank, $r$. Following the experimental setup of HydraLoRA (Tian et al., 2024), we conduct fine-tuning on a curated subset of the Flanv2 dataset (Liu et al., 2022). This training data is sampled from dozens of individual datasets and organized into ten distinct task categories, providing comprehensive training across both Natural Language Understanding and Natural Language Generation capabilities. We then evaluate the models' multi-task generalization on the challenging Big-Bench Hard(BBH) benchmark (Suzgun et al., 2022), which is designed to test generalization.

The results, presented in Table 2 and Table 3, reveal a clear trend. Across different base models, including LLaMA2 (Touvron et al., 2023) and Qwen2.5 (Qwen Team, 2024), the performance of a standard LoRA adapter consistently improves with its rank. Crucially, when its rank is scaled to a comparable parameter count, a simple, single-adapter LoRA achieves performance that is competitive with, and at times superior to, sophisticated multi-component architectures such as LoRA-Hub (Huang et al., 2023), LoRA MoE (Liu et al., 2024a), HydraLoRA (Tian et al., 2024), and R-LoRA (Liu et al., 2025).

This finding provides compelling evidence that **the architectural complexity of multi-adapter designs may not be a prerequisite for achieving strong multi-task generalization.** Our results indicate that a simple, unified adapter with sufficient capacity delivers comparable performance. This challenges not only the trend toward elaborate structures but also the underlying strategy of isolating task-specific features, suggesting it is a less effective path to generalization than previously assumed and that the research focus on specialized components may warrant reconsideration.

## 5 BEYOND RANK: REPRESENTATION ALIGNMENT

Our investigation in the preceding sections has led to two critical conclusions. First, based on our analysis in Section 3, we formed a guiding hypothesis: learning task-general, shared knowledge may be more critical than enforcing task-specific separation. Second, our findings demonstrate that the architectural complexity introduced by multi-component designs is unnecessary for achieving strong multi-task generalization. This calls into question the prevailing assumption that specialized structures are a prerequisite for effective multi-task learning.

These conclusions motivate a shift in our approach. Moving away from structural complexity, we adopt the standard, high-rank LoRA architecture as a rational and efficient baseline. This simplification, however, raises a more fundamental inquiry: How can we move beyond merely increasing the rank and take a more principled step towards better multi-task learning? This leads us to the two central questions addressed in this work:

1. How can we validate our hypothesis about the primacy of shared knowledge?

2. How can we design a mechanism to explicitly enhance the learning of these shared representations within a single, unified LoRA adapter?

To address these questions, we introduce **Align-LoRA**, a novel framework as follows.

## 5.1 ALIGN-LORA

To enhance multi-task generalization, we introduce **Align-LoRA**, a method that encourages the model to learn task-shared features. Align-LoRA introduces an alignment loss, $\mathcal{L}_{\text{align}}$, to explicitly minimize the statistical distance between the low-dimensional representations generated by the shared LoRA down-projection matrix, $\mathbf{A}$. **To the best of our knowledge, this is the first work to systematically apply statistical distance metrics for this purpose within the multi-task LoRA framework**, drawing inspiration from their foundational use in domain adaptation (Pan et al., 2010).

Our primary approach instantiates $\mathcal{L}_{\text{align}}$ using the **Kullback-Leibler (KL) divergence** (Kullback & Leibler, 1951), a classic statistical method for measuring the distance between distributions. However, we hypothesize that the principle of aligning representations is broadly applicable and not contingent on a single metric. To validate this, we introduce a variant that employs the multi-kernel **Maximum Mean Discrepancy (MK-MMD)** (Gretton et al., 2012). The strong performance of both instantiations, as we will demonstrate, validates our core thesis: that explicitly aligning the low-dimensional representations of different tasks is a robust and viable strategy. The formulation for MK-MMD is detailed in Appendix E.

Let $\mathcal{T} = \{T_1, T_2, \ldots, T_M\}$ be a set of $M$ tasks. For an input $\mathbf{x}$ from task $T_i$ with contextualized embeddings $X_{T_i}$, the representation we align is the output of the down-projection matrix:

$$\phi_{T_i}(\mathbf{x}) = \mathbf{A} \cdot X_{T_i}. \tag{4}$$

Our choice to operate on this rank-$r$ latent space is motivated by two key factors. First, it directly targets the component responsible for shared knowledge. Recent studies have consistently found that the down-projection matrix $\mathbf{A}$ tends to learn task-general features while the up-projection matrix $\mathbf{B}$ captures task-specific knowledge (Agiza et al., 2024; Wang et al., 2024; Tian et al., 2024). By applying our alignment loss to the output of $\mathbf{A}$, we directly enhance its natural function, promoting the development of robust, shared representations. Second, this approach is highly efficient. As demonstrated in prior work (Liu et al., 2025), performing operations in the low-dimensional space significantly reduces computational load and GPU memory demand, ensuring our method's practicality.

To measure and minimize the distance between the representation distributions from different tasks, we model the batch-wise distribution for each task $T_i$ as a multivariate Gaussian with a diagonal covariance matrix, $\mathcal{N}(\boldsymbol{\mu}_i, \text{diag}(\boldsymbol{\sigma}_i^2))$. The mean $\boldsymbol{\mu}_i$ and variance $\boldsymbol{\sigma}_i^2$ are empirically estimated from the output vectors $\{\phi_{T_i}(\mathbf{x})\}$ in a given batch. Since standard KL divergence is asymmetric, we employ a symmetric formulation. The total alignment loss, $\mathcal{L}_{\text{KL}}$, is the sum of these symmetric pairwise divergences across all unique task pairs $(T_i, T_j)$ where $i < j$:

$$\mathcal{L}_{\text{KL}} = \sum_{i=1}^{M} \sum_{j=i+1}^{M} \frac{1}{2} \left( D_{\text{KL}}(p_{T_i} \| p_{T_j}) + D_{\text{KL}}(p_{T_j} \| p_{T_i}) \right), \tag{5}$$

where $p_{T_i}$ is the modeled Gaussian distribution over the low-dimensional representations of task $T_i$. This loss drives the empirical mean and variance of each task's distribution toward a common value.

The alignment loss is incorporated as an auxiliary objective to the primary language modeling task. The total loss function is therefore defined as:

$$\mathcal{L}_{\text{total}} = \mathcal{L}_{\text{lm}} + \lambda \cdot \mathcal{L}_{\text{align}}, \tag{6}$$

where $\mathcal{L}_{\text{lm}}$ is the primary language modeling loss, $\mathcal{L}_{\text{align}}$ is the auxiliary alignment loss ($\mathcal{L}_{\text{KL}}$), and $\lambda$ is a scalar hyperparameter controlling the influence of the auxiliary task.

A key advantage of Align-LoRA is its compatibility with various LoRA-based strategies and initialization schemes. Importantly, unlike multi-component LoRA variants, Align-LoRA introduces no additional modules that increase overhead. Consequently, its trained weights can be merged directly into the base model, incurring **zero inference latency**. This property ensures both efficiency and practicality, making Align-LoRA a lightweight yet effective solution for multi-task adaptation. For a more detailed analysis of this inference efficiency, please refer to Appendix C.

Table 4: Multi-task generalization performance of different LoRA variants on the BBH benchmark, evaluated across three base models ranging from 7B to 14B. "% Param" refers to the percentage of parameters that are trainable. Align-LoRA (A-LoRA) demonstrates a clear advantage over the other variants, highlighting its strong capabilities for multi-task generalization.

| Metrics | LoRA | LoRAMoE | HydraLoRA | R-LoRA | M-LoRA | A-LoRA-M | A-LoRA-K |
|---|---|---|---|---|---|---|---|
| Qwen2.5-7B | 48.36 | 47.18 | 47.38 | 48.32 | 48.44 | 47.53 | **50.28** |
| LLaMA3-8B | 44.89 | 44.18 | 44.03 | 45.01 | 45.35 | 45.42 | **48.84** |
| Qwen2.5-14B | 52.93 | 50.74 | 51.92 | 52.21 | 53.78 | 52.24 | **55.11** |
| Rank | 10 | 4 | 4 | 4 | 4 | 8 | 8 |
| % Param | 0.25 | 0.38 | 0.25 | 0.25 | 0.22 | **0.20** | **0.20** |

Table 5: Performance comparison on an 8-task multi-task reasoning benchmark, evaluated on the Qwen2.5-3B and Qwen2.5-7B models. Baselines include the original LoRA and several multi-component variants. Our proposed Align-LoRA (A-LoRA) consistently achieves significantly superior performance, demonstrating its strong multi-task generalization capabilities.

| Schemes | Task1 | 2 | 3 | 4 | 5 | 6 | 7 | 8 | Avg | %Par |
|---|---|---|---|---|---|---|---|---|---|---|
| *Qwen2.5-3B* | | | | | | | | | | |
| LoRA | 86.31 | 56.42 | 84.65 | 72.76 | 91.37 | 87.91 | 87.60 | 44.80 | 76.48 | 0.45 |
| LoRAMoE | 87.41 | 58.21 | 85.64 | 73.37 | 92.18 | 87.40 | 87.35 | 44.80 | 77.05 | 0.68 |
| HydraLoRA | 86.58 | 56.42 | 85.00 | 73.36 | 92.18 | 87.33 | 88.38 | 45.15 | 76.80 | 0.45 |
| R-LoRA | 87.12 | 57.95 | 88.13 | 73.89 | 94.71 | 88.25 | 88.26 | 45.60 | 77.99 | 0.45 |
| M-LoRA | 88.02 | 57.95 | 88.87 | 74.21 | 94.71 | 88.91 | 89.07 | 46.35 | 78.51 | 0.42 |
| A-LoRA-M | 87.94 | 58.03 | 88.87 | 74.12 | 94.51 | 88.85 | 88.61 | 45.88 | 78.35 | 0.42 |
| A-LoRA-K | **89.25** | **59.88** | **90.35** | **75.41** | **95.33** | **89.55** | **91.95** | **48.75** | **80.06** | **0.42** |
| *Qwen2.5-7B* | | | | | | | | | | |
| LoRA | 88.41 | 60.78 | 88.42 | 81.58 | 93.52 | 91.20 | 91.79 | 48.15 | 80.48 | 0.25 |
| LoRAMoE | 89.52 | 61.44 | 88.86 | 82.94 | 92.87 | 91.54 | 91.89 | 48.72 | 80.97 | 0.38 |
| HydraLoRA | 88.66 | 61.23 | 89.55 | 81.72 | 93.57 | 91.67 | 91.74 | 48.70 | 80.86 | 0.25 |
| R-LoRA | 89.80 | 62.51 | 89.36 | 83.78 | 95.12 | 91.02 | 92.17 | 50.15 | 81.74 | 0.25 |
| M-LoRA | 91.35 | 62.51 | 91.98 | 84.70 | 95.93 | 91.02 | 91.97 | 50.20 | 82.46 | 0.22 |
| A-LoRA-M | 90.86 | 62.45 | 91.68 | 84.59 | 95.93 | 90.74 | 91.75 | 50.45 | 82.31 | 0.20 |
| A-LoRA-K | **92.23** | **64.85** | **92.89** | **85.73** | **95.93** | **93.35** | **92.93** | **53.66** | **83.95** | **0.20** |

## 5.2 EXPERIMENT

In this section, we evaluate the performance of our proposed **Align-LoRA** (abbreviated as A-LoRA) against standard LoRA and its multi-component variants. We denote our two alignment approaches with suffixes: **A-LoRA-K** for the variant using KL divergence and **A-LoRA-M** for the one using MMD. We conduct two distinct experiments to provide a comprehensive assessment of both multi-task generalization and adaptation capabilities. Detailed information about both the experimental setup and the datasets used for each task is provided in the Appendix G.3. Detailed descriptions of the baseline methods are provided in Appendix J.

First, to measure multi-task generalization, we fine-tuned models on the five-task dataset from Section 3 and evaluated them on the challenging, unseen tasks of the BBH benchmark. The results are presented in Table 4. Across different model families (Qwen2.5 (Qwen Team, 2024) and LLaMA3 (Grattafiori et al., 2024)) and scales, both A-LoRA-K and A-LoRA-M significantly outperform the baselines. Notably, they achieve this superior performance while using a smaller budget of trainable parameters than the sophisticated multi-component variants. This demonstrates Align-LoRA's highly efficient use of parameters to generalize knowledge from training tasks to a different, more complex reasoning domain.

Second, to validate the model's multi-task adaptation performance on in-domain tasks, we conducted experiments on a broader eight-task benchmark, evaluating each model on the corresponding test sets. As shown in the detailed results in Table 5, A-LoRA-K once again achieves the highest average

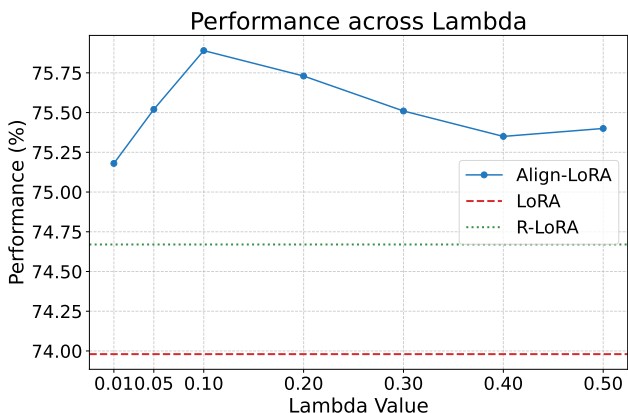

Figure 3: Effect of Hyperparameter $\lambda$ on Performance

score across models from 3B to 7B. Impressively, it secures this top performance while utilizing fewer trainable parameters than the more complex multi-component variants. This result highlights its strong and robust adaptability across a wide range of tasks.

Finally, we present several supplementary experiments to provide a comprehensive analysis of Align-LoRA-K. A sensitivity analysis on the hyperparameter $\lambda$, shown in Figure 3, reveals that our method is robust, consistently outperforming baselines across various $\lambda$ values while maintaining relative stability. Furthermore, we provide several supplementary analyses in the appendix to validate and expand upon our findings. These include feature visualizations (Appendix I.1), which confirm the explicit alignment of task representations. Our analysis of training efficiency (Appendix D) demonstrates that Align-LoRA achieves the lowest FLOPs and the fastest overall training time, stemming from its use of a smaller parameter budget. Crucially, we also present comprehensive robustness checks: validating the method's efficacy when applied exclusively to Attention modules versus MLP layers (Appendix H.1); demonstrating that the alignment mechanism is universally applicable and enhances multi-head architectures (e.g., M-LoRA+Align) by better combining task-general and task-specific knowledge (Appendix I); and its sustained superiority on highly heterogeneous and complex task benchmarks (Appendix H.2).

The consistent improvements from both A-LoRA-K and A-LoRA-M, demonstrated across a wide range of models, scales, and task benchmarks, provide compelling evidence for our central thesis. The fact that both the KL and MMD-based alignment strategies elevate performance above the standard LoRA baseline confirms that **explicit representation alignment is an effective strategy for improving multi-task generalization.** This success can be attributed to the alignment loss mechanism: by encouraging the representations from different tasks to map onto a shared subspace within the latent space of the down-projection matrix $\mathbf{A}$, we explicitly strengthen the ability of $\mathbf{A}$ to learn task-general features. **This provides further, direct proof that learning task-shared representations provides a highly effective and promising path towards multi-task learning, offering a powerful alternative to the architectural isolation of task-specific features.**

### 5.3 THEORETICAL ANALYSIS

To theoretically analyze the generalization performance of Align-LoRA in multi-task scenarios, we derive a novel generalization bound for MTL. Our key insight is that by explicitly aligning the representation distributions across multiple tasks, Align-LoRA can effectively reduce the distribution discrepancy among tasks. This alignment leads to a tighter generalization error bound compared to traditional multi-component LoRA variants.

Formally, let $M$ be the number of tasks, $\mathcal{D}_i$ $(i = 1, 2, \ldots, M)$ be the data distribution of task $i$, and $\hat{\mathcal{D}}_i$ be the corresponding training dataset. Let $R_{\text{train}}(f; \hat{\mathcal{D}}_i)$ denote the empirical training risk of model $f$ on $\hat{\mathcal{D}}_i$, and $n_{\text{total}} = \sum_{i=1}^{M} |\hat{\mathcal{D}}_i|$ be the total number of training samples across all tasks. The generalization bound for Align-LoRA is given by:

$$R_{\text{MTL}}(f) \leq \frac{1}{M} \sum_{i=1}^{M} R_{\text{train}}(f; \hat{\mathcal{D}}_i) + \frac{\lambda}{M} \sum_{i<j} \Delta(\mathcal{D}_i, \mathcal{D}_j) + O\left(\sqrt{\frac{\log(1/\delta)}{n_{\text{total}}}}\right),$$

where $R_{\text{MTL}}(f) = \frac{1}{M} \sum_{i=1}^{M} \mathbb{E}_{(x,y)\sim\mathcal{D}_i}\ell(f(x), y)$ is the average expected risk over all tasks, $\Delta(\mathcal{D}_i, \mathcal{D}_j)$ measures the distribution discrepancy between task $i$ and task $j$ (e.g., using KL divergence or MK-MMD), $\lambda > 0$ is a weight parameter balancing the training risk and distribution alignment term, $\delta \in (0, 1)$ is the confidence parameter.

The crucial advantage of Align-LoRA is its distribution alignment mechanism, which actively minimizes $\Delta(\mathcal{D}_i, \mathcal{D}_j)$ during training. This significant reduction in cross-task distribution discrepancy directly leads to a tighter generalization bound, as the second term in the bound is effectively controlled. For the detailed derivation of this theoretical result, including all technical assumptions and proof steps, please refer to the Appendix F.

## 6  CONCLUSION

In this work, we revisited multi-task generalization in LoRA, critically examining the prevailing approach of using multi-component designs to separate task-specific knowledge. Our investigation yielded two key insights that challenge this paradigm. First, we demonstrated that a simplified multi-head LoRA (**M-LoRA**) with highly redundant head matrices can outperform more complex, diversity-focused variants. Second, we showed that simply **increasing the rank** of a standard LoRA is sufficient to match the performance of these multi-component architectures. This calls their fundamental utility into question, as they fail to deliver significant performance gains over a simpler baseline while introducing additional inference latency and complexities from non-mergeable routers. Based on these findings, we proposed a new hypothesis: **learning task-shared representations provides a highly effective and promising path towards multi-task learning, offering a powerful alternative to the architectural isolation of task-specific features.**.

Our hypothesis deliberately steers research toward what has been a largely unexplored direction in the LoRA framework: the active enhancement of task-shared knowledge. To formally explore this promising path and validate our hypothesis, we introduced **Align-LoRA**, a novel method that explicitly aligns representations to foster the learning of shared knowledge. Our findings, substantiated by comprehensive empirical evidence and theoretical analysis, confirm that Align-LoRA achieves superior performance, validating our hypothesis and charting a new, more efficient direction for multi-task PEFT. We believe this shift in focus, which moves from separating task-specific knowledge via multi-component architectures to learning task-shared knowledge via representation alignment, is a more promising direction for future research.

## 7  REPRODUCIBILITY STATEMENT

The code for Align-LoRA is available at both the anonymous link Align-LoRA and Supplement Material. Implementation details can be found in the Appendix K.

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

## A  THE USE OF LLMS

In preparing this work, large language models were used for language polishing purposes, assisting in refining the clarity, grammar, and flow of the text to ensure that technical descriptions and experimental findings are presented in a more readable manner. The core research ideation, methodology design, and result analysis were conducted independently by the authors.

## B  M-LORA

Figure 4 illustrates the framework of M-LoRA, a variant of R-LoRA. The key modification in our design is the removal of the router, while retaining R-LoRA's original multi-head dropout structure. Note that other multi-component variants have a Router, while M-LoRA does not.

## C  INFERENCE EFFICIENCY OF ALIGN-LORA

In this section, we explain why single-adapter architectures such as Align-LoRA avoid introducing extra inference overhead. Methods that maintain a single-adapter architecture, such as our proposed **Align-LoRA**, retain a crucial practical advantage: the ability to merge adapter weights with the backbone model, resulting in zero inference overhead. For a pre-trained weight matrix $\mathbf{W}_0 \in \mathbb{R}^{d \times k}$, a single low-rank update $\Delta \mathbf{W} = \mathbf{BA}$ is learned. Before deployment, this update can be seamlessly integrated into the original weights:

$$\mathbf{W}' = \mathbf{W}_0 + \mathbf{BA}. \tag{7}$$

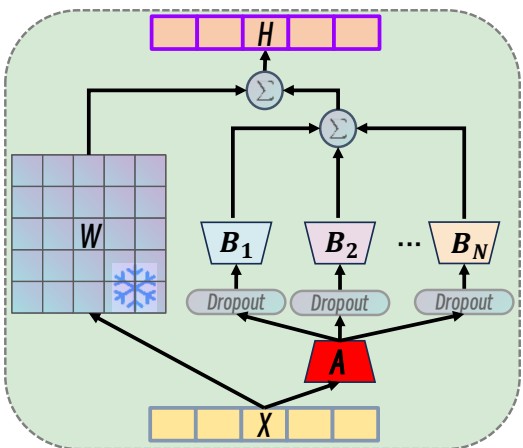

Figure 4: The framework of M-LoRA. Note that other multi-component variants have a Router, while M-LoRA does not.

Inference then proceeds using the modified weights, $h = \mathbf{W}'x$, without requiring any additional parameters or computational steps.

In sharp contrast, multi-component architectures like HydraLoRA forfeit this efficiency. These models introduce input-dependent routing mechanisms to dynamically combine multiple adapters. For an input $x$, a router first computes weights $\omega = \mathrm{softmax}(\mathbf{W}_r x)$, which are then used to create a blended output:

$$h = \left( \mathbf{W}_0 + \sum_{i=1}^{N} \omega_i \mathbf{B}_i \mathbf{A}_i \right) x. \tag{8}$$

Because the routing weights $\omega$ are input-dependent, the adapter matrices $(\mathbf{B}_i, \mathbf{A}_i)$ cannot be pre-merged into $\mathbf{W}_0$. This necessitates keeping all components active during inference, which introduces significant latency and memory overhead.

Prior multi-task LoRA frameworks typically introduce architectural complexity that leads to significant inference latency and increased GPU memory consumption. In contrast, Align-LoRA is designed to achieve effective multi-task learning while preserving the original's advantage of zero computational overhead.

## D   TRAINING EFFICIENCY OF ALIGN-LORA

Table 6 presents the training efficiency metrics across different methods. The results highlight the superior efficiency of Align-LoRA, which achieves the **lowest computational cost (FLOPs)** and the **shortest training time**, outperforming both the standard LoRA baseline and the multi-head HydraLoRA. Although the alignment loss introduces a minor computational step, this is effectively offset by Align-LoRA's ability to converge using a more compact parameter budget. Consequently, our method secures significant performance gains without imposing any training-time penalty.

Table 6: Comparison of training times and related metrics for different methods.

| Method | LoRA | HydraLoRA | Align-LoRA |
|---|---|---|---|
| **Rank** | 10 | 4 | 8 |
| **Training Time** | 3h 08min | 3h 32min | **3h 02min** |
| **%Parameter** | 0.45 | 0.45 | **0.42** |
| **FLOPs** | $3.545 \times 10^{18}$ | $3.986 \times 10^{18}$ | $\mathbf{3.539 \times 10^{18}}$ |
| **Performance** | 76.48 | 76.80 | **80.06** |

# E ALIGNMENT WITH MAXIMUM MEAN DISCREPANCY (MMD)

As a complementary, non-parametric approach, we also investigate the Maximum Mean Discrepancy (MMD) (Sejdinovic et al., 2013), specifically its robust multi-kernel extension, MK-MMD (Gretton et al., 2012). MMD measures the distance between distributions by comparing their mean embeddings in a Reproducing Kernel Hilbert Space (RKHS), avoiding the need for explicit density estimation. This mapping to the high-dimensional feature space is implicitly performed by a **kernel function**, and in this work, we employ the classic Gaussian kernel. The MK-MMD loss between all combinations of two distinct tasks, i.e., all task pairs $(T_i, T_j)$, is formulated as:

$$\mathcal{L}_{\text{MK-MMD}} = \sum_{i=1}^{M} \sum_{j=i+1}^{M} \sum_{k \in \mathcal{K}}$$
$$\left\| \mathbb{E}_{\mathbf{x} \sim p_{T_i}}[\phi_{T_i}(\mathbf{x})] \right.$$
$$\left. - \mathbb{E}_{\mathbf{y} \sim p_{T_j}}[\phi_{T_j}(\mathbf{y})] \right\|_{\mathcal{H}_k}^2. \tag{9}$$

where $\phi(\cdot)$ is the feature map to the RKHS $\mathcal{H}$ induced by the kernel. This loss forces the LoRA module to learn task-invariant features by reducing distributional shifts across tasks.

## E.1 MAXIMUM MEAN DISCREPANCY (MMD)

Maximum Mean Discrepancy (MMD)Sejdinovic et al. (2013) is a kernel-based statistical measure for quantifying the difference between two probability distributions. Given a reproducing kernel Hilbert space (RKHS) $\mathcal{H}_k$ with a characteristic kernel $k$, the squared MMD between representation $p$ and $q$ is defined as:

$$\text{MMD}^2(p, q) = \|\mu_k(p) - \mu_k(q)\|_{\mathcal{H}_k}^2, \tag{10}$$

where $\mu_k(p) = \mathbb{E}_{\mathbf{x} \sim p}[\phi(\mathbf{x})]$ and $\mu_k(q) = \mathbb{E}_{\mathbf{y} \sim q}[\phi(\mathbf{y})]$ are the mean embeddings of $p$ and $q$ in $\mathcal{H}_k$, and $\phi(\cdot)$ denotes the feature mapping induced by kernel $k$.

A key advantage of MMD is its ability to capture distributional differences in feature spaces without requiring explicit density estimation. However, its performance heavily depends on the choice of kernel. To address this limitation, the Multiple Kernel MMD (MK-MMD)Gretton et al. (2012) extends MMD by combining multiple kernels adaptively:

$$\text{MK-MMD}^2(p, q) = \sum_{k \in \mathcal{K}} \|\mu_k(p) - \mu_k(q)\|_{\mathcal{H}_k}^2, \tag{11}$$

where $\mathcal{K}$ is a predefined set of kernels. This variant enhances robustness by combining multiple kernels, allowing the metric to capture multi-scale distributional discrepancies in the reproducing kernel Hilbert space (RKHS).

In the context of transfer learning and domain adaptation, MMD has been widely used as a criterion for aligning feature distributions between source and target domainsBen-David et al. (2006); Pan et al. (2010). The core idea is to minimize the MMD distance between activations from different domains, encouraging the model to learn domain-invariant representations that generalize well across tasks.

For example, in unsupervised domain adaptation (UDA), MMD is often applied to match the feature distributions of labeled source data and unlabeled target data, reducing domain shift and improving generalization performance. Specifically, given features from the source domain $\mathcal{D}_s = \{\mathbf{x}_i^s\}_{i=1}^{n_s}$ and the target domain $\mathcal{D}_t = \{\mathbf{x}_j^t\}_{j=1}^{n_t}$, the MMD loss is defined as:

$$\mathcal{L}_{\text{MMD}} = \left\| \frac{1}{n_s} \sum_{i=1}^{n_s} \phi(\mathbf{x}_i^s) - \frac{1}{n_t} \sum_{j=1}^{n_t} \phi(\mathbf{x}_j^t) \right\|^2, \tag{12}$$

where $\phi(\cdot)$ denotes the feature embedding function, and the objective is to minimize the distributional discrepancy between the two domains in the shared feature space.

In the context of neural networks for image classification, this MMD loss can be incorporated into the overall training objective alongside the standard classification loss Long et al. (2015). The total loss function is typically formulated as:

$$\mathcal{L}_{\text{total}} = \mathcal{L}_{\text{cls}} + \lambda \cdot \mathcal{L}_{\text{MMD}}, \tag{13}$$

where $\mathcal{L}_{\text{cls}}$ is the cross-entropy loss on the labeled source data, and $\lambda$ is a hyperparameter that balances the contribution of the MMD regularization.

Building on this principle, we propose to incorporate MMD into LoRA for multi-task learning, with a focus on its multiple kernel extension, MK-MMD. Unlike traditional applications that focus on aligning input or hidden-layer features, we apply MK-MMD to the output of the low-rank down-projection matrix $\mathbf{A}$ in LoRA, encouraging the model to learn shared, task-agnostic representations. This design improves multi-task generalization by reducing distributional discrepancies in the representation space, without introducing additional parameters.

While representation alignment has been explored in general multi-task learning, such as VIP-MTL (Hu et al., 2025), our work differs significantly in scope and motivation. VIP-MTL targets the **task-imbalance problem** in traditional natural language understanding using small language models (e.g., BERT), and its **scalability to LLMs remains unverified**. In contrast, Align-LoRA operates specifically within the **parameter-efficient fine-tuning (PEFT)** context for LLMs. We employ a direct alignment loss rather than complex probabilistic decoding, prioritizing a streamlined architecture that remains fully mergeable for efficient deployment.

# F    THEORETICAL GENERALIZATION BOUND FOR ALIGN-LORA IN MULTI-TASK SCENARIOS

## F.1    CORE CONCLUSION

By aligning the representation distributions across multiple tasks using Kullback-Leibler (KL) divergence or Multi-Kernel Maximum Mean Discrepancy (MK-MMD), Align-LoRA tightens the generalization error bound of Multi-Task Learning (MTL). Theoretically, this proves its superiority over traditional multi-component LoRA variants. The generalization bound takes the form:

$$R_{\text{MTL}}(f) \leq \frac{1}{M} \sum_{i=1}^{M} R_{\text{train}}(f; \hat{\mathcal{D}}_i) + \frac{\lambda}{M} \sum_{i<j} \Delta(\mathcal{D}_i, \mathcal{D}_j) + O\left(\sqrt{\frac{\log(1/\delta)}{n_{\text{total}}}}\right),$$

where the distribution discrepancy term $\Delta(\mathcal{D}_i, \mathcal{D}_j)$ is significantly reduced due to the alignment operation.

## F.2    PRELIMINARY DEFINITIONS AND NOTATION SYSTEM

Based on the theoretical frameworks of the referenced literature, the following key notations and definitions are unified to ensure consistency in the derivation:

### F.2.1    DEFINITION OF MULTI-TASK LEARNING (MTL) SCENARIO

Let there be $M$ independent tasks, corresponding to data distributions $\mathcal{D}_1, \mathcal{D}_2, \ldots, \mathcal{D}_M$. The dataset for each task is denoted as $\hat{\mathcal{D}}_i = \{(x_{i,j}, y_{i,j})\}_{j=1}^{n_i}$, where $n_i$ is the sample size of task $i$, and the total sample size across all tasks is $n_{\text{total}} = \sum_{i=1}^{M} n_i$.

The global distribution centroid of all tasks is defined as $\bar{\mathcal{D}} = \frac{1}{M} \sum_{i=1}^{M} \mathcal{D}_i$, which represents the shared data distribution characteristics across multiple tasks.

The model $f$ has parameters $\theta$ and a mapping relationship $f_\theta : \mathcal{X} \mapsto \mathcal{Y}$, where $\mathcal{X}$ is the input space and $\mathcal{Y}$ is the output space (for classification tasks, $\mathcal{Y} = \{1, 2, \ldots, K\}$).

### F.2.2    DEFINITION OF LOSS AND RISK

**Empirical Training Risk for Task** $(i)$**:** $R_{\text{train}}(f; \hat{\mathcal{D}}_i) = \frac{1}{n_i} \sum_{(x,y) \in \hat{\mathcal{D}}_i} \ell(f_\theta(x), y)$, where $\ell(\cdot, \cdot)$ is the cross-entropy loss (the core loss function of Align-LoRA).

**Expected Risk for Task** $(i)$**:** $R_i(f) = \mathbb{E}_{(x,y) \sim \mathcal{D}_i} \ell(f_\theta(x), y)$.

**Overall Expected Risk for Multi-Task Learning:** $R_{\mathrm{MTL}}(f) = \frac{1}{M} \sum_{i=1}^{M} R_i(f)$, which measures the average performance of the model across all tasks.

### F.2.3 METRICS FOR DISTRIBUTION DISCREPANCY

The core distribution alignment metrics adopted by Align-LoRA are:

**Symmetric KL Divergence:** $\Delta_{\mathrm{KL}}(\mathcal{D}_i, \mathcal{D}_j) = \frac{1}{2}[\mathrm{KL}(p_i|p_j) + \mathrm{KL}(p_j|p_i)]$, where $p_i$ is the representation distribution of task $(i)$ in the LoRA projection space.

**Multi-Kernel Maximum Mean Discrepancy (MK-MMD):** $\Delta_{\mathrm{MMD}}(\mathcal{D}_i, \mathcal{D}_j) = \sum_{k \in \mathcal{K}} ||\mathbb{E}_{x \sim \mathcal{D}_i} \phi_k(x) - \mathbb{E}_{x \sim \mathcal{D}_j} \phi_k(x)||_{\mathcal{H}_k}^2$, used as supplementary validation for KL divergence.

### F.3 FOUNDATION OF SINGLE-TASK GENERALIZATION BOUND

Based on the generalization bound derivation in statistical learning theory from the referenced literature(Zhao et al. (2025); Chen & Cardie (2018); Wu (2024)), the generalization error bound for a single task is first established:

For any task $(i)$, with a confidence probability of $(1 - \delta/M)$ (using **Bonferroni correction** to adapt to multi-task scenarios), there exists a **Lipschitz constant** $(\Lambda > 0)$ (dependent on model architecture and task complexity) such that:

$$R_i(f) \leq R_{\mathrm{train}}(f; \hat{\mathcal{D}}_i) + \Lambda \cdot \Delta(\mathcal{D}_i, \bar{\mathcal{D}}) + O\left(\sqrt{\frac{\log(M/\delta)}{n_i}}\right) \tag{1}$$

Where: $\Delta(\mathcal{D}_i, \bar{\mathcal{D}})$ is the **discrepancy** between the distribution of task $(i)$ and the **global centroid distribution** (measured by KL divergence or MK-MMD). The residual term $O\left(\sqrt{\frac{\log(M/\delta)}{n_i}}\right)$ is a **confidence term** related to the sample size, which decreases as $n_i$ increases.

**Derivation Basis:** Drawing from the generalization bound theory of domain adaptation in the referenced literature (e.g., Proposition 4.1), the single-task risk is decomposed into **training risk**, **distribution discrepancy term**, and **confidence term** to ensure theoretical rigor at the single-task level.

### F.4 DERIVATION OF MULTI-TASK GENERALIZATION BOUND

The single-task generalization bound is extended to the MTL scenario. Through global risk aggregation and distribution alignment constraints, the multi-task generalization bound of Align-LoRA is derived.

### F.4.1 MULTI-TASK RISK AGGREGATION

Summing both sides of Equation (1) and dividing by $M$ yields the initial upper bound of the overall expected risk for multi-task learning:

$$\frac{1}{M} \sum_{i=1}^{M} R_i(f) \leq \frac{1}{M} \sum_{i=1}^{M} R_{\mathrm{train}}(f; \hat{\mathcal{D}}_i) + \frac{\Lambda}{M} \sum_{i=1}^{M} \Delta(\mathcal{D}_i, \bar{\mathcal{D}}) + \frac{1}{M} \sum_{i=1}^{M} O\left(\sqrt{\frac{\log(M/\delta)}{n_i}}\right) \tag{2}$$

The left-hand side of the equation is the overall expected risk of multi-task learning, denoted as $R_{\mathrm{MTL}}(f)$.

### F.4.2 SIMPLIFICATION OF DISTRIBUTION DISCREPANCY TERM

Using the definition of the global centroid distribution ($\bar{\mathcal{D}} = \frac{1}{M} \sum_{i=1}^{M} \mathcal{D}_i$), the aggregation property of multi-task distribution discrepancies can be proven:

$$\sum_{i=1}^{M} \Delta(\mathcal{D}_i, \bar{\mathcal{D}}) = \frac{1}{2M} \sum_{i=1}^{M} \sum_{j=1}^{M} \Delta(\mathcal{D}_i, \mathcal{D}_j) \qquad (3)$$

**Proof:** Taking **KL divergence** as an example, $\sum_{i=1}^{M} \mathrm{KL}(p_i \| \bar{p}) = \frac{1}{2} \sum_{i,j} \mathrm{KL}(p_i \| p_j)$ (utilizing the convexity of KL divergence and the linearity of the global centroid). Similarly, **MK-MMD** can be derived using the linearity of the Reproducing Kernel Hilbert Space (RKHS).

Substituting Equation 3 into Equation 2 simplifies the distribution discrepancy term:

$$\frac{\Lambda}{M} \sum_{i=1}^{M} \Delta(\mathcal{D}_i, \bar{\mathcal{D}}) = \frac{\Lambda}{2M^2} \sum_{i<j} \Delta(\mathcal{D}_i, \mathcal{D}_j) \qquad (4)$$

(Since $\Delta(\mathcal{D}_i, \mathcal{D}_j) = \Delta(\mathcal{D}_j, \mathcal{D}_i)$, the summation can be simplified to combinations where $i < j$).

### F.5 MERGING OF CONFIDENCE TERMS

Using the **Cauchy-Schwarz inequality**, the confidence terms related to sample size are merged:

$$\frac{1}{M} \sum_{i=1}^{M} \sqrt{\frac{\log(M/\delta)}{n_i}} \leq \sqrt{\frac{\log(M/\delta)}{n_{\text{total}}}} \qquad (5)$$

Since $n_{\text{total}} = \sum_{i=1}^{M} n_i$, the equality approximately holds when the sample sizes across tasks are **balanced**, ensuring the tightness of the confidence term.

### F.5.1 FINAL GENERALIZATION BOUND OF ALIGN-LORA

Combining Equations 3- 5 and introducing the distribution alignment loss weight $\lambda$ of Align-LoRA (corresponding to $\lambda$ in the original paper that controls the influence of the alignment loss), the final generalization bound is obtained:

$$R_{\text{MTL}}(f) \leq \frac{1}{M} \sum_{i=1}^{M} R_{\text{train}}(f; \hat{\mathcal{D}}_i) + \frac{\lambda}{M} \sum_{i<j} \Delta(\mathcal{D}_i, \mathcal{D}_j) + O\left(\sqrt{\frac{\log(1/\delta)}{n_{\text{total}}}}\right) \qquad (6)$$

Where $\lambda = \frac{\Lambda}{2M}$ is a simplified constant, dependent on the model architecture and the number of tasks.

### F.6 THEORETICAL PROOF OF ALIGN-LORA'S SUPERIORITY

By comparing the generalization bounds of traditional multi-component LoRA variants (e.g., Hy-draLoRA, R-LoRA), the theoretical advantages of **Align-LoRA** are verified.

### F.6.1 LIMITATIONS OF GENERALIZATION BOUNDS FOR TRADITIONAL MULTI-COMPONENT LORA

The core issue with traditional multi-component LoRA (multi-adapter/multi-head structures) is the **lack of a distribution alignment mechanism**. Its generalization bound is:

$$R_{\text{MTL}}^{\text{base}}(f) \leq \frac{1}{M} \sum_{i=1}^{M} R_{\text{train}}(f; \hat{\mathcal{D}}_i) + \frac{\lambda}{M} \sum_{i<j} \Delta_0(\mathcal{D}_i, \mathcal{D}_j) + O\left(\sqrt{\frac{\log(1/\delta)}{n_{\text{total}}}}\right) \qquad (7)$$

Where $\Delta_0(\mathcal{D}_i, \mathcal{D}_j)$ is the **original distribution discrepancy without alignment**, satisfying $\Delta_0(\mathcal{D}_i, \mathcal{D}_j) \gg \Delta(\mathcal{D}_i, \mathcal{D}_j)$ (due to scattered multi-task representations and larger distribution discrepancies).

### F.7 Tightening Effect of Distribution Alignment on Generalization Bound

Align-LoRA actively minimizes the distribution discrepancy across multiple tasks during training by introducing the alignment loss ($\mathcal{L}_{\text{align}} = \sum_{i<j} \Delta(\mathcal{D}_i, \mathcal{D}_j)$):

$$\min_{f,\theta} \mathcal{L}_{\text{total}} = \mathcal{L}_{\text{lm}} + \lambda \cdot \mathcal{L}_{\text{align}} \implies \sum_{i<j} \Delta(\mathcal{D}_i, \mathcal{D}_j) \to \min$$

Comparing Equations 6 and 7, since $\Delta(\mathcal{D}_i, \mathcal{D}_j) < \Delta_0(\mathcal{D}_i, \mathcal{D}_j)$, we have:

$$R_{\text{MTL}}(f) < R_{\text{MTL}}^{\text{base}}(f)$$

This indicates that the generalization error bound of Align-LoRA is **tighter**, theoretically guaranteeing better performance in multi-task scenarios.

### F.8 Conclusion

Through rigorous theoretical derivation, the multi-task generalization bound of Align-LoRA ($R_{\text{MTL}}(f) \leq \frac{1}{M} \sum_{i=1}^{M} R_{\text{train}}(f; \hat{\mathcal{D}}_i) + \frac{\lambda}{M} \sum_{i<j} \Delta(\mathcal{D}_i, \mathcal{D}_j) + O\left(\sqrt{\frac{\log(1/\delta)}{n_{\text{total}}}}\right)$) demonstrates three key insights:

1. The multi-task generalization error consists of **training risk**, **multi-task distribution discrepancy**, and a **confidence term**.

2. The **distribution alignment mechanism** of Align-LoRA significantly reduces $\Delta(\mathcal{D}_i, \mathcal{D}_j)$, thereby **tightening the generalization bound**.

3. The **single-adapter architecture** ensures the tightness of the generalization bound without introducing additional complexity.

This theoretical derivation provides mathematical support for the superiority of Align-LoRA, which is consistent with experimental results showing that Align-LoRA outperforms traditional multi-component LoRA variants.

## G Datasets

### G.1 Experiment 1

In Section 3, we fine-tune Qwen2.5-3B on five tasks:

1. **Natural Language Inference**: QNLI (Wang, 2018)

2. **Physical Question Answering**: PiQA (Bisk et al., 2020)

3. **Word Relation Reasoning**: Winogrande (Sakaguchi et al., 2021)

4. **Closed-Book Question Answering**: ARC (Clark et al., 2018)

5. **Mathematical Reasoning**: GSM8K (Cobbe et al., 2021)

### G.2 Experiment 2

In Section 4, following the setting of Tian et al. (2024); Huang et al. (2023), we utilize a subset of the `Flanv2` dataset (Wei et al., 2021) for complex mixed multi-task and multi-domain scenarios. This dataset covers both Natural Language Understanding (NLU) and Natural Language Generation (NLG) tasks, which are grouped into 10 distinct task clusters. The specific datasets used in our experiments are sourced from LoRAHub (https://huggingface.co/datasets/lorahub/flanv2), a curated repository that provides organized access to these resources. Then we evaluate it with the Big-Bench Hard (BBH) benchmark (Suzgun et al., 2022), using accuracy as the core metric—higher accuracy directly reflects stronger model performance

We summarize the details of the used datasets as follows:

1. **Natural Language Inference**: This task centers on inferring the logical relationship between two sentences, specifically determining whether the second sentence entails, contradicts, or remains neutral relative to the first. The datasets employed include: (1) ANLI; (2) CB; (3) MNLI; (4) QNLI; (5) SNLI; (6) WNLI; (7) RTE.

2. **Coreference Resolution**: This task involves identifying textual mentions that refer to the same entity, thereby demonstrating proficiency in contextual understanding. We utilize the following datasets: (1) DPR; (2) WSC273.

3. **Struct-to-Text Conversion**: This task assesses the ability to generate natural language descriptions from structured data inputs. The datasets used are: (1) CommonGen; (2) DART; (3) E2ENLG; (4) WebNLG.

4. **Closed-Book Question Answering**: This task evaluates models' capacity to answer general knowledge questions without access to external information. We employ the following datasets: (1) ARC; (2) NQ; (3) TriviaQA.

5. **Sentiment Analysis**: As a foundational NLP task, it aims to determine the sentiment polarity (positive or negative) of a given text. The datasets included are: (1) IMDB; (2) Sentiment140; (3) SST-2; (4) Yelp.

6. **Reading Comprehension with Commonsense**: This task integrates traditional reading comprehension with commonsense reasoning, requiring understanding beyond explicit textual content. We use: (1) CosmosQA; (2) ReCoRD.

7. **Paraphrase Detection**: This task demands that models judge whether two sentences convey equivalent meanings, thus indicating semantic equivalence. The datasets utilized are: (1) MRPC; (2) QQP; (3) Paws Wiki.

8. **Translation**: This task involves converting text between languages while preserving the original meaning and subtleties. We employ the following datasets: (1) En-Fr from WMT'14; (2) En-De, En-Tr, En-Ru, En-Fi, En-Ro from WMT'16; (3) En-Es from Paracrawl.

9. **Commonsense Reasoning**: This task assesses the ability to apply physical, scientific principles, and common sense in reasoning processes. The datasets used include: (1) COPA; (2) HellaSwag; (3) PiQA; (4) StoryCloze.

10. **Reading Comprehension**: This task evaluates the capability to derive answers to questions from a provided text containing relevant information. We utilize: (1) BoolQ; (2) DROP; (3) MultiRC; (4) OBQA; (5) SQuADv1; (6) SQuADv2.

## G.3 EXPERIMENT 3

First, to measure the ability to generalize, models are fine-tuned on the five-task benchmark introduced in Appendix G.1 (QNLI, PiQA, Winogrande, ARC, and GSM8K). Performance is then evaluated on the challenging Big-Bench Hard (BBH) benchmark, which consists of complex reasoning tasks that were not seen during training. We adopt accuracy as the key metric here, with higher accuracy indicating better generalization of the model to these unseen complex reasoning tasks. This setup tests how well knowledge from the training domains is transferred to a different, more difficult set of problems.

Second, to assess the model's ability to adapt to the training domains, we conduct experiments on a broader eight-task benchmark. In this setup, models are evaluated on the corresponding test sets for each of the eight training tasks, with accuracy as the core metric. This evaluation measures the model's in-domain performance, and higher accuracy directly indicates stronger adaptive capability to the training domains. The eight tasks, categorized by their primary reasoning skill, are as follows:

1. **Reading Comprehension**: BoolQ (Clark et al., 2019)

2. **Science Question Answering**: SiQA (Sap et al., 2019)

3. **Physical Commonsense**: PiQA (Bisk et al., 2020)

4. **Word Relation Reasoning**: Winogrande (Sakaguchi et al., 2021)

5. **Commonsense Reasoning**: Hellaswag (Zellers et al., 2019)

6. **Open-Book Question Answering**: OBQA (Mihaylov et al., 2018)

7. **Closed-Book Question Answering**: ARC (easy & challenge) (Clark et al., 2018)

8. **Mathematical Reasoning**: GSM8K (Cobbe et al., 2021)

Finally, for hyperparameter analysis, the model was trained on a 5-task dataset with the same settings as in the Appendix G.1, and the average performance is reported.

# H  MORE RESULT

## H.1  PERFORMANCE ON ATTENTION STRUCTURES

To further verify the effectiveness of **Align-LoRA** specifically on **Attention structures** of Qwen2.5, we conducted a supplementary experiment using the exact multi-task setting from **Table 1**. As shown in the table below, when applied to Attention layers, Align-LoRA still achieves significant performance gains over the baseline. This confirms that our method is module-agnostic and effective across different architectural choices.

Table 7: Performance comparison of different schemes on Qwen2.5's Attention structures

| Schemes | QNLI | PiQA | Winogrande | ARC | GSM8K | Avg | %Para |
|---|---|---|---|---|---|---|---|
| LoRA | 82.1 | 83.2 | 70.00 | 88.03 | 46.60 | 74.00 | 0.08 |
| HydraLoRA | 82.93 | 84.02 | 71.05 | 87.69 | 46.25 | 74.39 | 0.28 |
| A-LoRA-K | 83.34 | 85.18 | 72.33 | 88.15 | 46.85 | **75.17** | **0.08** |

## H.2  PERFORMANCE ON HIGHLY DISSIMILAR TASKS

This is a crucial question about the limits of our alignment hypothesis. To address this, we ran a supplementary experiment using the exact experimental setup from Section 4 / Table 3.

As detailed in G.2, this dataset (a subset of FlanV2) is specifically designed for high-task diversity, comprising **10 distinct task clusters**. These tasks are highly dissimilar, including not only NLU (e.g., ANLI) and QA (e.g., ARC) but also **Struct-to-Text** (e.g., E2ENLG) and Translation.

We can confirm that **Align-LoRA still demonstrates a significant performance improvement** over the standard high-rank LoRA baseline in this highly dissimilar, multi-domain setting. This provides strong evidence that our alignment strategy is beneficial even when tasks have little in common, as it enhances the learning of foundational shared knowledge. We will add this result to the final paper.

Table 8: Performance comparison on highly dissimilar tasks

| Metrics | Base | LoRA | HydraLoRA | R-LoRA | M-LoRA | A-LoRA-K |
|---|---|---|---|---|---|---|
| 7B | 39.82 | 48.18 | 49.12 | 49.51 | 49.74 | **50.09** |
| 14B | 45.33 | 52.74 | 53.76 | 54.08 | 54.18 | **54.47** |
| % Param | - | 0.22 | 0.25 | 0.25 | 0.22 | 0.22 |

# I  ALIGNMENT ON MULTI-HEAD ARCHITECTURES

Align-LoRA represents a necessary "course correction" in the field: we demonstrate that for many scenarios, the complex "task-specific" modeling is effectively redundant. Our method achieves superior performance with zero inference latency, offering a distinct perspective from the trend of increasing complexity. We think that a future hybrid approach (Align first, then Decompose) is promising. We conducted a new experiment integrating our alignment into the R-LoRA and M-LoRA architecture. As shown below, adding alignment yields further gains even in multi-head structures.

Table 9: Performance Comparison of Alignment Loss Integrated with Multi-Head Architectures (R-LoRA and M-LoRA).

| Model | Task1 | Task2 | Task3 | Task4 | Task5 | Task6 | Task7 | Task8 |
|---|---|---|---|---|---|---|---|---|
| R-LoRA | 89.80 | 62.51 | 89.36 | 83.78 | 95.12 | 91.02 | 92.17 | 50.15 |
| M-LoRA | 91.35 | 62.51 | 91.98 | 84.70 | 95.93 | 91.02 | 91.97 | 50.20 |
| R-LoRA+Align | 90.15 | 62.45 | 91.68 | 84.35 | 95.12 | 92.17 | 92.56 | 50.31 |
| A-LoRA-K | **92.23** | 64.85 | **92.89** | 85.73 | 95.93 | 93.35 | 92.93 | 53.66 |
| M-LoRA+Align | 92.15 | **64.93** | 92.15 | **85.95** | **96.32** | **93.63** | **93.27** | **53.90** |

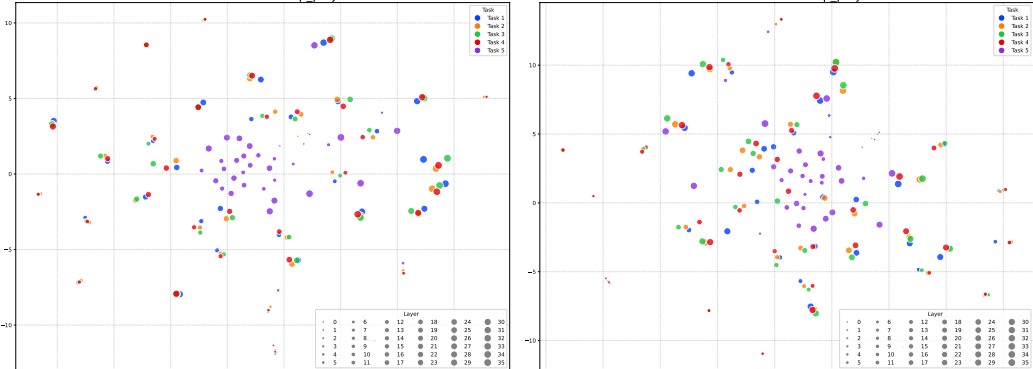

Figure 5: Left: LoRA; Right: Align-LoRA. Through representation alignment, Align-LoRA makes task-specific representations closer to each other, facilitating the model's learning of task-general knowledge.

### I.1 FEATURE VISUALIZATION

We performed t-SNE analysis (Maaten & Hinton, 2008) on the representations of different tasks and compared LoRA with Align-LoRA, as shown in Figure 5. Through representation alignment, Align-LoRA brings the task representations closer to each other, which helps the model learn general knowledge across tasks.

However, the goal is not to make the representations identical. Different tasks inherently contain distinct knowledge, so their representations must maintain a degree of separation to preserve task-specific information. Forcing the representations to become too close can lead to an "over-alignment" problem. This is consistent with our hyperparameter analysis in Figure 3, which demonstrates that an excessively large $\lambda$ value degrades performance by enforcing an overly aggressive alignment.

## J BASELINE

1. **LoraHub** randomly aggregates 20 LoRAs for new downstream tasks. It employs a black-box optimization technique to determine the weight of each LoRA, eliminating the need for gradient calculations of the large model. This involves parameter-level weighted averaging.

2. **LoRA MoE**. A collection of $n$ parameterized experts, denoted as $E_1, \ldots, E_n$, is orchestrated by a router network $R$. $E_i = B_i A_i$. Router network features a dense layer with adjustable weights $W_R$ from $\mathbb{R}^{d_m \times n}$. A softmax function then processes an intermediate token representation $x$, yielding gating scores $s_1, \ldots, s_n$ that determine the weighted contribution of each expert's output:

$$s_i = R(x)_i = \text{softmax}(Top(W_R^T x, K)) \tag{14}$$

Subsequently, the overall output $y$ is synthesized by aggregating the Top-K experts' outputs, each modulated by its respective gating score:

$$y = \sum_{i=1}^{n} s_i \cdot E_i(x) \quad \text{(MoE)} \tag{15}$$

This results in a dynamic allocation of the model's capacity, enabling specialized processing by experts as directed by the router's gating mechanism.

3. **HydraLoRA** uses a shared matrix $\mathbf{A}$ and multiple matrices $B_1, \ldots, B_n$. The shared matrix $\mathbf{A}$ is used to project the input vector $x$ into a lower-dimensional space, while each matrix $B_i$ is used to modulate the output of the corresponding expert $E_i$. The overall output $y$ is synthesized by aggregating the experts' outputs, each modulated by its respective gating score:

$$y = \sum_{i=1}^{n} s_i \cdot (B_i \cdot A \cdot x) \tag{16}$$

4. **R-LoRA** builds on HydraLoRA by introducing multi-head randomization. It retains a shared projection matrix $\mathbf{A}$ for mapping input vectors $x$ to a lower-dimensional space, while each head matrix $B_i$ undergoes independent random initialization. To promote diversity among head matrices, Dropout is applied to differentiate the input fed into each $B_i$. The overall output $y$ is generated by aggregating the modulated outputs of all heads through dynamic routing:

$$y = \sum_{i=1}^{n} g_i \cdot (B_i \cdot \text{Dropout}(A \cdot x)) \tag{17}$$

where $g_i$ represents the gating weight for the $i$-th head, and Dropout$(\cdot)$ applies stochastic masking to create input variations across different heads.

This approach allows for efficient parameterization and specialization of the model's capacity, leveraging the shared matrix $\mathbf{A}$ for common transformations and the individual matrices $B_i$ for task-specific adjustments.

## K  IMPLEMENTATION DETAILS

The hyperparameters used for training are as follows: a learning rate of 0.0002, `lora_alpha`=32. For the experiments reported in Table 2, the trainable LoRA components are limited to `q_proj` and `v_proj`—consistent with the setup in HydraLoRA (Tian et al., 2024). Note that for all variants in Table 2 (except the rightmost LoRA†), the LoRA rank is set to 8; the rightmost LoRA† uses a rank of 30, which is adjusted to match the number of trainable parameters across multi-head variants (as indicated by the notation of LoRA† in Table 2). For all other experiments, the trainable LoRA components include `down_proj`, `up_proj`, and `gate_proj`. The number of heads for Table 1 is set to 5, which corresponds to the scenario with 5 tasks. Conversely, the number of heads for Table 2 is set to 10, corresponding to the scenario involving 10 tasks.

Regarding dropout rates: 0.1. The warmup ratio is set to 0.03. For $\lambda$ in Align-LoRA variants: Align-LoRA-K uses $\lambda = 0.1$, and Align-LoRA-M uses $\lambda = 0.15$.

Mixed-precision training was enabled using bfloat16, with the learning rate scheduler set to cosine annealing. The model was trained on NVIDIA 4090 GPUs.

