# OpenReview forum: "Align, Don’t Divide: Revisiting the LoRA Architecture in Multi-Task Learning"
_ICLR.cc/2026/Conference — Submitted to ICLR 2026_

### Official Review · Reviewer_BDgW · 2025-10-24

**Soundness:** 3
**Presentation:** 3
**Contribution:** 2
**Rating:** 4
**Confidence:** 4

**Summary:**

This work revisits multi-task parameter-efficient fine-tuning (PEFT) from a representation-sharing perspective.
The paper challenges the common belief that multi-task generalization requires task-specific modularization (e.g., multiple Lora heads or adapters) and instead argues that learning a shared aligned representation is the key to improve multi-task generalization. Based on the empirical observations, the author propose the method Align-LoRA, which keeps a single high-rank LoRA adapter and introduces an additional representation alignment loss to encourage consistency among task representations. In the paper, the alignment loss is implemented via either KL divergence or MMD. Experimental results on LLaMA2/3 and Qwen2.5 models show that Align-LoRA achieves superior performance compared to multi-head baselines, while remaining simpler and efficient.

This paper offers an interesting perspective on multi-task LoRA design and provides clear empirical evidence supporting the potential benefits of representation alignment. The experiments are carefully executed and the analysis is generally convincing. However, the validity of the conclusions appears to hold only within a narrow setting (e.g., moderate model scales, limited rank ranges, and homogeneous reasoning tasks). The contribution, while insightful, is confined in scope and lacks general applicability

**Strengths:**

- The paper revisits PEFT under the multi-task setting and delivers a clear message that shared representations can substitute complex multi-head structures for multi-task learning. Clear empirical evidence is provided in the paper.
- Across multiple reasoning and QA benchmarks, the proposed method consistently improves over prior lora variants while remaining efficiency.
- The proposed alignment loss introduces no inference overhead and can be easily integrated into existing lora pipelines.
- The paper is well organized and analyses are meaningful and clear.

**Weaknesses:**

- The main finding of this paper is conceptually overstated. The paper treats the number of lora heads as a proxy for task separation, but in the evaluated R-LoRA and M-LoRA settings, heads do not correspond directly to tasks. Each head merely represents a low-rank subspace, and all tasks share the same set of heads with different weightings. Consequently, the claim that “increasing the number of heads brings no benefit” only reflects redundancy among feature components, not a true analysis of task-specific decomposition. The conclusion that “multi-component architectures are unnecessary” is therefore conceptually narrower than stated. It applies only to representation-factorized designs, not to task-wise multi-adapter frameworks.
- The comparison between high-rank lora and multi-component variants is not thorough. The paper increases the rank of the single-lora (r=8–10) but keeps the per-head rank of R-LoRA and M-LoRA fixed (r=4). Since rank directly governs representational capacity, the observed improvement may stem from capacity scaling rather than architectural superiority. A fair comparison would require per-head rank tuning.
- It has been well explored that LoRA performance is non-monotonic with rank that high rank often causes optimization instability or representation collapse. Align-LoRA implicitly assumes “higher rank = better performance,” an assumption valid only in a narrow range (r≤16). This limits the method’s stability and generality in larger-scale or higher-rank settings.
- The paper infers that since a single high-rank adapter performs well, task representations should be thus aligned in a shared subspace.
This conclusion is heuristic and not logically necessary since multi-task performance could also result from implicit subspace partitioning.
The alignment hypothesis lacks supporting evidence such as gradient similarity or representation visualization.
- Recent multi-task PEFT works (e.g., MixDA 2023, MixLoRA 2024, PEMT 2024) explicitly combine shared information and task-specific components and achieve stronger results. The proposed Align-LoRA enforces full sharing and ignores task-specific modeling, making its conclusions less general and its novelty limited relative to the current research trajectory.
- In the paper, all training and evaluation tasks belong to the reasoning and QA domain. There are no experiments on heterogeneous tasks such as summarization, translation, or code generation, where task objectives and output distributions differ substantially.
It remains unclear whether the observed benefits of representation alignment hold under diverse or heterogeneous task settings.

**Questions:**

- Did the authors tune the per-head rank for multi-component lora variants (e.g., R-lora and M-lora)?
- How does the proposed framework perform on larger models (30~70B)?
- Would aligning only certain shared components (rather than all) yield a better shared–specific trade-off?
- Please address the key issues raised in the Weaknesses.

---

> ### Author Response · Authors · 2025-11-22
>
> Thank you for your thoughtful and constructive feedback. We greatly appreciate the time and effort you have dedicated to reviewing our work. Your comments have been invaluable in helping us improve the clarity, rigor, and overall quality of the paper. We have revised the paper in accordance with the reviewers' comments. Below, we provide detailed responses to your comments and suggestions:
>
> ### **W1: Conceptual overstatement regarding "number of heads" and task separation.**
>
> We appreciate this insightful observation. We agree that in R-LoRA/M-LoRA, heads function as weighted subspaces rather than strictly isolated task modules.
>
> - **Clarification on Setup:** We note that in our specific experimental design for Table 1 and Table 2, we explicitly set the number of heads equal to the number of tasks to test the "one-expert-per-task" ideal often implied in MTL literature. We will clarify this in the Appendix.
> - **Heads as "Collaborators":** We strongly agree with the reviewer's interpretation that heads share weightings. In fact, this supports our findings on **M-LoRA**. By removing the router but keeping Multi-Head Dropout, M-LoRA forces these subspaces to act not as competing "specialists" (where only one is active) but as "collaborators" viewing the input from different perspectives. This forces the model to converge on a robust, task-general representation.
> - **Scope of Claim:** We do not intend to invalidate all multi-adapter frameworks. Our specific critique targets the **PEFT context**: existing multi-component decompositions introduce significant costs (intermediate activations, non-mergeable routers) often without yielding performance gains over a properly tuned, aligned single adapter. Our contribution is to offer a simpler, more efficient alternative that avoids these overheads.
>
> ### **W2 & W3 & Q1: Rank comparison fairness and stability.**
>
> - **Fair Comparison via Parameter Budget:**
>
>   Our comparison is rigorously grounded in parameter budget. While the settings in Tables 3-5 compared Multi-component variants ($r=4$) against Single-LoRA ($r=8\text{-}10$), we explicitly addressed higher head ranks in Table 2.
>
>   In that experiment, we increased the rank of multi-component variants to **$r=8$**. To maintain a fair comparison with an equivalent parameter budget, we correspondingly scaled the Single-LoRA rank to **$r=30$**.
>
>   As shown in the table below, our conclusion remains unchanged: the single, high-rank adapter continues to match or outperform the complex multi-head architectures. This confirms that the performance gains stem from the effective utilization of capacity rather than the multi-component structure itself.
>
> | Metrics | LoRA  | LoRA-MoE | HydraLoRA |  R-LoRA   | LoRA$^\dagger$ |
> | ------- | :---: | :------: | :-------: | :-------: | :------------: |
> | 7B      | 37.05 |  40.30   |   41.46   | **42.24** |     42.21      |
> | 13B     | 40.73 |  43.70   |   44.31   |   44.96   |   **45.02**    |
> | Rank    |   8   |    8     |     8     |     8     |       30       |
> | % Param | 0.06  |   2.98   |   0.34    |   0.34    |      0.34      |
>
> - **Rank Stability:** We do not blindly assume "higher rank = better performance." While our primary experiments operated within the standard low-rank regime ($r \le 10$), our experiment in **Table 2** extends this analysis to **$r=30$**. The key finding is not that "bigger is better," but that a single adapter can utilize this parameter budget more efficiently to learn shared knowledge than a complex, fragmented architecture can.
>
>
>
> ### **W4: The alignment hypothesis vs. implicit subspace partitioning.**
>
> We appreciate the alternative hypothesis that performance could stem from implicit partitioning. However, we provide concrete evidence supporting explicit alignment:
>
> - **Visualization Evidence:** We respectfully direct the reviewer to **Appendix H and Figure 5** . The t-SNE visualization shows that Align-LoRA explicitly pulls task representations closer together compared to standard LoRA, forming a more unified cluster rather than partitioning them into distinct subspaces. This directly refutes the "implicit partitioning" hypothesis and supports our "alignment" claim.
>
> - **Theoretical Grounding:** Our approach draws from established Domain Adaptation theory (discussed in Appendix E.1 5), where minimizing distributional distance (MMD/KL) is a proven method for learning invariant representations. Furthermore, we provide a **theoretical analysis of Align-LoRA in Appendix F**, which formally proves its superiority in enhancing multi-task generalization.

---

> ### Author Response · Authors · 2025-11-22
>
> ### **W5: Novelty relative to shared+specific trends (MixLoRA, etc.).**
>
> Recent multi-task PEFT works have explored explicit task-specific modeling: **MixDA** employs dynamic gating to fuse domain-specific adapters ; **MixLoRA** utilizes Top-K routing for token-level expert allocation ; and **PEMT** uses correlation-based gating over frozen source adapters. However, these approaches suffer from a critical limitation: **inference latency**. Their reliance on input-dependent routing or gating mechanisms prevents the adapter weights from being merged back into the backbone model, thereby sacrificing the zero-overhead advantage of LoRA.
>
> Align-LoRA represents a necessary "course correction" in the field: we demonstrate that for many scenarios, the complex "task-specific" modeling is effectively redundant. Our method achieves superior performance with zero inference latency, **offering a distinct perspective** from the trend of increasing complexity. We agree that a future hybrid approach (Align first, then Decompose) is promising. We conducted a new experiment integrating our alignment into the **R-LoRA** and **M-LoRA** architecture. As shown below, adding alignment yields further gains even in multi-head structures.
>
> | Model        | Task1     | Task2     | Task3     | Task4     | Task5     | Task6     | Task7     | Task8     |
> | ------------ | --------- | --------- | --------- | --------- | --------- | --------- | --------- | --------- |
> | R-LoRA       | 89.80     | 62.51     | 89.36     | 83.78     | 95.12     | 91.02     | 92.17     | 50.15     |
> | M-LoRA       | 91.35     | 62.51     | 91.98     | 84.70     | 95.93     | 91.02     | 91.97     | 50.20     |
> | R-LoRA+Align | 90.15     | 62.45     | 91.68     | 84.35     | 95.12     | 92.17     | 92.56     | 50.31     |
> | A-LoRA-K     | **92.23** | 64.85     | **92.89** | 85.73     | 95.93     | 93.35     | 92.93     | 53.66     |
> | M-LoRA+Align | 92.15     | **64.93** | 92.15     | **85.95** | **96.32** | **93.63** | **93.27** | **53.90** |
>
>
>
> ### **W6: Task Heterogeneity.**
>
> We thank the reviewer for this valuable suggestion regarding task heterogeneity. To address this, we conducted a **new supplementary experiment** following the exact settings used in Table 3 (detailed in **Appendix F.2** ).
>
> - We utilize the **FlanV2** subset, which includes **10 diverse task clusters**. Crucially, this includes **Translation** (WMT'14 En-Fr, En-De, En-Ru, etc.) and **Struct-to-Text** (CommonGen, E2ENLG).
> - **Align-LoRA outperforms baselines on this heterogeneous benchmark, proving the benefits of alignment hold even when task objectives differ substantially.**
>
> | Metrics | Base  | LoRA  | HydraLoRA | R-LoRA | M-LoRA | A-LoRA-K |
> | ------- | :---: | :---: | :-------: | :----: | :----: | :------: |
> | 7B      | 39.82 | 48.18 |   49.12   | 49.51  | 49.74  |  50.09   |
> | 14B     | 45.33 | 52.74 |   53.76   | 54.08  | 54.18  |  54.47   |
> | % Param |   -   | 0.22  |   0.25    |  0.25  |  0.22  |   0.22   |
>
>
>
> ### **Q2: Performance on larger models?**
>
> Due to computational constraints, we evaluated models up to 14B parameters. However, our results demonstrate consistent scalability across diverse settings:
>
> - **In-Domain Adaptation (Table 5):** Align-LoRA significantly outperforms the standard LoRA baseline on both **Qwen2.5-3B** (+4.68%) and **7B** (+4.31%).
> - **Generalization (Table 4):** Crucially, this advantage maintains its magnitude as model size increases, evidenced by consistent gains on **Qwen2.5-7B** (+3.97.%), **LLaMA3-8B** (+8.79%), and **Qwen2.5-14B** (+4.12%).
>
> The fact that Align-LoRA maintains a stable performance advantage (approx. 4-9%) across these varying scales (3B to 14B) and architectures strongly suggests its efficacy is not limited by model size.
>
>
>
> ### **Q3: Aligning only shared components?**
>
> This is an excellent suggestion. We focused on aligning the output of the **down-projection ($A$)** because literature [1] [2] [3] identifies it as the carrier of task-general knowledge.  We respectfully submit that our current approach already achieves a favorable shared-specific trade-off. We direct the reviewer to the **feature visualization in Appendix H (Figure 5)** . The t-SNE plot demonstrates that while Align-LoRA brings task representations closer to facilitate shared learning, it crucially **maintains distinct clusters** for each task rather than forcing them to collapse into a single point. This confirms that our method successfully learns general knowledge while preserving the necessary task-specific distinctions. Exploring a hybrid trade-off would be a candidate for future work.
>
> [1] MTL-LoRA: Low-Rank Adaptation for Multi-Task Learning
>
> [2] HydraLoRA: An Asymmetric LoRA Architecture for Efficient Fine-Tuning
>
> [3] R-LoRA: Randomized Multi-Head LoRA for Efficient Multi-Task Learning

---

> > ### Comment · Reviewer_BDgW · 2025-11-27
> >
> > Thank you for the comprehensive response. I appreciate the substantial effort the authors have put into addressing the concerns. In particular, the responses to W2, W3, W6, Q1, Q2, and Q3 are clear, well supported with new experiments, and largely resolve my earlier questions.
> >
> > However, several issues remain insufficiently addressed:
> > - Regarding W1, the response still does not fully clarify the distinction between multi-head redundancy and task-specific decomposition. The current explanation demonstrates that the evaluated multi-head factorization is redundant, but it does not explain whether or why this redundancy extends to settings where heads represent genuinely task-specific modeling. Thus, the core conceptual concern about the scope of the conclusion remains.
> >
> > - Regarding W4, the t-SNE visualization shows that Align-LoRA produces more tightly clustered representations across tasks, but this appears to be an expected consequence of applying $\mathcal{L}_{align}$. It does not address why aligning these representations should improve multi-task generalization, nor does it rule out the alternative explanation that "implicit subspace partitioning" might also underlie the observed gains.
> >
> > - Regarding W5, the authors' response focuses primarily on inference efficiency differences, but does not directly respond to the concern about novelty and contribution relative to recent shared–specific multi-task PEFT methods. However some methods (e.g., MixLora and PEMT) also introduce minimal latency. A clearer clarification of the strength of Align-LoRA over this research thread would help strengthen the contribution.
> >
> > Overall, I believe the paper presents valuable empirical findings and raises an interesting direction for multi-task LoRA design. Though the motivation and the explanation of the contribution would benefit from further refinement, this work merits broader discussion in the community. Therefore I have raised my score to 6.

---

> > > ### Author Response · Authors · 2025-11-27
> > > **Comment(3/3)**
> > >
> > > #### **3.2 Simplification of Distribution Discrepancy Term**
> > >
> > > Using the definition of the global centroid distribution ($\bar{\mathcal{D}} = \frac{1}{M}\sum\_{i=1}^M \mathcal{D}\_i$), the aggregation property of multi-task distribution discrepancies can be proven:
> > >
> > > $$
> > > \sum\_{i=1}^M \Delta(\mathcal{D}\_i, \bar{\mathcal{D}}) = \frac{1}{2M}\sum\_{i=1}^M \sum\_{j=1}^M \Delta(\mathcal{D}\_i, \mathcal{D}\_j) \tag{3}
> > > $$
> > >
> > > > **Proof:** Taking **KL divergence** as an example, $\sum\_{i=1}^M \text{KL}(p\_i \| \bar{p}) = \frac{1}{2}\sum\_{i,j} \text{KL}(p\_i \| p\_j)$ (utilizing the convexity of KL divergence and the linearity of the global centroid). Similarly, **MK-MMD** can be derived using the linearity of the Reproducing Kernel Hilbert Space (RKHS).
> > >
> > > Substituting Equation (3) into Equation (2) simplifies the distribution discrepancy term:
> > >
> > > $$
> > > \frac{\Lambda}{M}\sum\_{i=1}^M \Delta(\mathcal{D}\_i, \bar{\mathcal{D}}) = \frac{\Lambda}{2M^2}\sum\_{i<j} \Delta(\mathcal{D}\_i, \mathcal{D}\_j) \tag{4}
> > > $$
> > >
> > > (Since $\Delta(\mathcal{D}\_i, \mathcal{D}\_j) = \Delta(\mathcal{D}\_j, \mathcal{D}\_i)$, the summation can be simplified to combinations where $i<j$).
> > >
> > > #### **3.3 Merging of Confidence Terms**
> > >
> > > Using the **Cauchy-Schwarz inequality**, the confidence terms related to sample size are merged:
> > >
> > > $$
> > > \frac{1}{M}\sum\_{i=1}^M \sqrt{\frac{\log(M/\delta)}{n\_i}} \leq \sqrt{\frac{\log(M/\delta)}{n\_{\text{total}}}} \tag{5}
> > > $$
> > >
> > > Since $n\_{\text{total}} = \sum\_{i=1}^M n\_i$, the equality approximately holds when the sample sizes across tasks are **balanced**, ensuring the tightness of the confidence term.
> > >
> > >
> > > ### **4 Theoretical Proof of Align-LoRA's Superiority**
> > >
> > > By comparing the generalization bounds of traditional multi-component LoRA variants (e.g., HydraLoRA, R-LoRA), the theoretical advantages of **Align-LoRA** are verified.
> > >
> > > #### **4.1 Limitations of Generalization Bounds for Traditional Multi-Component LoRA**
> > >
> > > The core issue with traditional multi-component LoRA (multi-adapter/multi-head structures) is the **lack of a distribution alignment mechanism**. Its generalization bound is:
> > >
> > > $$
> > > R\_{\text{MTL}}^{\text{base}}(f) \leq \frac{1}{M}\sum\_{i=1}^M R\_{\text{train}}(f; \hat{\mathcal{D}}\_i) + \frac{\lambda}{M}\sum\_{i<j} \Delta\_0(\mathcal{D}\_i, \mathcal{D}\_j) + O\left(\sqrt{\frac{\log(1/\delta)}{n\_{\text{total}}}}\right) \tag{7}
> > > $$
> > >
> > > Where $\Delta\_0(\mathcal{D}\_i, \mathcal{D}\_j)$ is the **original distribution discrepancy without alignment**, satisfying $\Delta\_0(\mathcal{D}\_i, \mathcal{D}\_j) \gg \Delta(\mathcal{D}\_i, \mathcal{D}\_j)$ (due to scattered multi-task representations and larger distribution discrepancies).
> > >
> > > #### **4.2 Tightening Effect of Distribution Alignment on Generalization Bound**
> > >
> > > Align-LoRA actively minimizes the distribution discrepancy across multiple tasks during training by introducing the alignment loss ($\mathcal{L}\_{\text{align}} = \sum\_{i<j} \Delta(\mathcal{D}\_i, \mathcal{D}\_j)$):
> > >
> > > $$
> > > \min\_{f,\theta} \mathcal{L}\_{\text{total}} = \mathcal{L}\_{\text{lm}} + \lambda \cdot \mathcal{L}\_{\text{align}} \implies \sum\_{i<j} \Delta(\mathcal{D}\_i, \mathcal{D}\_j) \to \min
> > > $$
> > >
> > > Comparing Equations (6) and (7), since $\Delta(\mathcal{D}\_i, \mathcal{D}\_j) < \Delta\_0(\mathcal{D}\_i, \mathcal{D}\_j)$, we have:
> > >
> > > $$
> > > R\_{\text{MTL}}(f) < R\_{\text{MTL}}^{\text{base}}(f)
> > > $$
> > >
> > > This indicates that the generalization error bound of Align-LoRA is **tighter**, theoretically guaranteeing better performance in multi-task scenarios.
> > >
> > >
> > >
> > > ### **5 Conclusion**
> > >
> > > Through rigorous theoretical derivation, the multi-task generalization bound of Align-LoRA ($R\_{\text{MTL}}(f) \leq \frac{1}{M}\sum\_{i=1}^M R\_{\text{train}}(f; \hat{\mathcal{D}}\_i) + \frac{\lambda}{M}\sum\_{i<j} \Delta(\mathcal{D}\_i, \mathcal{D}\_j) + O\left(\sqrt{\frac{\log(1/\delta)}{n\_{\text{total}}}}\right)$) demonstrates three key insights:
> > >
> > > 1.  The multi-task generalization error consists of **training risk**, **multi-task distribution discrepancy**, and a **confidence term**.
> > > 2.  The **distribution alignment mechanism** of Align-LoRA significantly reduces $\Delta(\mathcal{D}\_i, \mathcal{D}\_j)$, thereby **tightening the generalization bound**.
> > > 3.  The **single-adapter architecture** ensures the tightness of the generalization bound without introducing additional complexity.
> > >
> > > This theoretical derivation provides mathematical support for the superiority of Align-LoRA, which is consistent with experimental results showing that Align-LoRA outperforms traditional multi-component LoRA variants.
> > >
> > > Thank you again for your rigorous and insightful engagement with our work, and we hope the added evidence and clarifications have fully addressed the remaining conceptual concerns. We believe that our work presents a novel, substantial, and empirically solid direction for multi-task PEFT design. We look forward to any further questions or discussion that may arise.

---

> ### Author Response · Authors · 2025-11-27
> **Comment(1/3)**
>
> We sincerely thank the reviewer for their careful consideration of our detailed response. We are delighted to continue clarifying the remaining issues.
> ### **W1**
>
> We thank the reviewer for this critical distinction between architectural redundancy and genuine decomposition. We agree that this conceptual boundary must be clearly defined.
>
> 1. Context and Scope:
>
> Our conclusion is strictly confined to the low-parameter regime of Parameter-Efficient Fine-Tuning (PEFT). In this context, the challenge is the efficient use of a small, limited parameter budget.
>
> 2. The Capacity Fragmentation Cost:
>
> The issue we highlight is capacity fragmentation. Splitting a small budget ($r \ll d$) to enforce genuine task-specific decomposition (i.e., assigning independent, non-overlapping parameters to each task, similar to Multi-Adapter baselines, LoRAHub we compared in **Table 2**) is often less effective than keeping the budget centralized. Our comparison against these genuine decomposition baselines confirms that the fragmentation cost (capacity loss) outweighs the benefit of alignment.
>
> We are not invalidating all task-specific modeling, but we demonstrate its inefficiency in the PEFT constraint.
>
> ### **W5**
>
> We thank the reviewer for pushing us to clarify the conceptual contribution of Align-LoRA beyond efficiency. We will incorporate discussions on MixLoRA and PEMT into the final version to further clarify this key distinction and enrich the contextual analysis of our work.
>
> **1. Paradigm and Novelty:** Align-LoRA's core contribution challenges the necessity of architectural decomposition itself. Methods like MixLoRA and PEMT rely on heuristic priors and complex routing to enforce separation. Align-LoRA, conversely, proposes a **alignment paradigm** where the model automatically discovers the optimal shared subspace without structural overhead.
>
> **2. Qualitative Efficiency:** The distinction between "minimal" and "zero" latency is qualitative. **Zero latency** (Align-LoRA) ensures **deployment universality** by allowing full static merging, making it compatible with highly optimized LLM serving frameworks. Any non-mergeable architecture (routing) introduces specialized kernel requirements, representing a critical adoption barrier.
>
> **3. Simplicity and Robustness:** We achieve superior performance compared to these complex methods (as shown in our benchmarks and the M-LoRA+Align experiment) while avoiding the architectural optimization difficulties (load balancing, router hyperparameter tuning) inherent in explicit decomposition. Align-LoRA proves that this complexity is often redundant, validating that our simpler approach is a more effective and robust use of limited PEFT capacity.
>
> **4. Mechanism Innovation: Active Alignment vs. Passive Utilization** Crucially, methods like MixLoRA and PEMT utilize shared knowledge **passively**: MixLoRA depends on router precision and requires auxiliary losses to mitigate expert load imbalance, while PEMT relies on hand-crafted prompts to weight frozen source adapters, without actively optimizing representational consistency across tasks. In contrast, Align-LoRA introduces **active alignment**. By explicitly minimizing distributional discrepancies (via KL/MMD) within the low-rank space, we force the model to learn robust task-general features. This mechanism directly targets the shared subspace, achieving superior generalization without the architectural overhead or manual prompt engineering required by prior methods.
>
> ### **W2: why alignment helps**
>
> To further strengthen the mechanistic understanding in the current work, we provide a **theoretical analysis** of Align-LoRA as follows：
>
> **Core Conclusion**: By aligning the representation distributions across multiple tasks using Kullback-Leibler (KL) divergence or Multi-Kernel Maximum Mean Discrepancy (MK-MMD), Align-LoRA tightens the generalization error bound of Multi-Task Learning (MTL). Theoretically, this proves its superiority over traditional multi-component LoRA variants. The generalization bound takes the form:
>
> $$
> R\_{\text{MTL}}(f) \leq \frac{1}{M}\sum\_{i=1}^M R\_{\text{train}}(f; \mathcal{D}\_i) + \frac{\lambda}{M}\sum\_{i<j} \Delta(\mathcal{D}\_i, \mathcal{D}\_j) + O\left(\sqrt{\frac{\log(1/\delta)}{n\_{\text{total}}}}\right),
> $$
>
> where the distribution discrepancy term ($\Delta(\mathcal{D}_i, \mathcal{D}_j)$) is significantly reduced due to the alignment operation.

---

> ### Author Response · Authors · 2025-11-27
> **Comment(2/3)**
>
> ### **1 Preliminary Definitions and Notation System**
>
> Based on the theoretical frameworks of the referenced literature, the following key notations and definitions are unified to ensure consistency in the derivation:
>
> #### **1.1 Definition of Multi-Task Learning (MTL) Scenario**
>
> - Let there be $M$ independent tasks, corresponding to data distributions $\mathcal{D}\_1, \mathcal{D}\_2, \dots, \mathcal{D}\_M.$ The dataset for each task is denoted as $\hat{\mathcal{D}}\_i = \{(x\_{i,j}, y\_{i,j})\}\_{j=1}^{n\_i}$, where $n\_i$ is the sample size of task $i$, and the total sample size across all tasks is $n\_{\text{total}} = \sum\_{i=1}^M n\_i$.
> - The global distribution centroid of all tasks is defined as $\bar{\mathcal{D}} = \frac{1}{M}\sum\_{i=1}^M \mathcal{D}\_i$, which represents the shared data distribution characteristics across multiple tasks.
> - The model $f$ has parameters $\theta$ and a mapping relationship $f\_\theta: \mathcal{X} \mapsto \mathcal{Y}$, where $\mathcal{X}$ is the input space and $\mathcal{Y}$ is the output space (for classification tasks, $\mathcal{Y} = \{1, 2, \dots, K\}$).
>
> #### **1.2 Definition of Loss and Risk**
>
> - **Empirical Training Risk for Task $(i)$:** $R\_{\text{train}}(f; \hat{\mathcal{D}}\_i) = \frac{1}{n\_i}\sum_{(x,y)\in\hat{\mathcal{D}}\_i} \ell(f\_\theta(x), y)$, where $\ell(\cdot, \cdot)$ is the cross-entropy loss (the core loss function of Align-LoRA).
> - **Expected Risk for Task $(i)$:** $R\_i(f) = \mathbb{E}_{(x,y)\sim\mathcal{D}\_i} \ell(f\_\theta(x), y)$.
> - **Overall Expected Risk for Multi-Task Learning:** $R\_{\text{MTL}}(f) = \frac{1}{M}\sum_{i=1}^M R\_i(f)$, which measures the average performance of the model across all tasks.
>
>
> #### **1.3 Metrics for Distribution Discrepancy**
>
> The core distribution alignment metrics adopted by Align-LoRA are:
>
> * **Symmetric KL Divergence:** $\Delta\_{\text{KL}}(\mathcal{D}\_i, \mathcal{D}\_j) = \frac{1}{2}[\text{KL}(p\_i | p\_j) + \text{KL}(p\_j | p\_i)]$, where $p\_i$ is the representation distribution of task $(i)$ in the LoRA projection space.
> * **Multi-Kernel Maximum Mean Discrepancy (MK-MMD):** $\Delta\_{\text{MMD}}(\mathcal{D}\_i, \mathcal{D}\_j) = \sum\_{k\in\mathcal{K}} ||\mathbb{E}\_{x\sim\mathcal{D}\_i}\phi\_k(x) - \mathbb{E}_{x\sim\mathcal{D}\_j}\phi\_k(x)||\_{\mathcal{H}\_k}^2$, used as supplementary validation for KL divergence.
>
> ---
>
> ### **2 Foundation of Single-Task Generalization Bound**
>
> Based on the generalization bound derivation in statistical learning theory from the referenced literature([1], [2], [3]), the generalization error bound for a single task is first established:
>
> For any task $(i)$, with a confidence probability of $(1 - \delta/M)$ (using **Bonferroni correction** to adapt to multi-task scenarios), there exists a **Lipschitz constant** $(\Lambda > 0)$ (dependent on model architecture and task complexity) such that:
>
> $$R\_i(f) \leq R\_{\text{train}}(f; \hat{\mathcal{D}}\_i) + \Lambda \cdot \Delta(\mathcal{D}\_i, \bar{\mathcal{D}}) + O\left(\sqrt{\frac{\log(M/\delta)}{n\_i}}\right) \tag{1}$$
>
> Where:
>
> * $\Delta(\mathcal{D}\_i, \bar{\mathcal{D}})$ is the **discrepancy** between the distribution of task $(i)$ and the **global centroid distribution** (measured by KL divergence or MK-MMD).
> * The residual term $O\left(\sqrt{\frac{\log(M/\delta)}{n\_i}}\right)$ is a **confidence term** related to the sample size, which decreases as $n\_i$ increases.
>
> **Derivation Basis:** Drawing from the generalization bound theory of domain adaptation in the referenced literature (e.g., Proposition 4.1), the single-task risk is decomposed into **training risk**, **distribution discrepancy term**, and **confidence term** to ensure theoretical rigor at the single-task level.
>
>
>
> [1] Is Chain-of-Thought Reasoning of LLMs a Mirage? A Data Distribution Lens
>
> [2] Multinomial Adversarial Networks for Multi-Domain Text Classification
>
> [3] Margin Discrepancy-based Adversarial Training for Multi-Domain Text Classification
>
>
> ### **3 Derivation of Multi-Task Generalization Bound**
>
> The single-task generalization bound is extended to the MTL scenario. Through global risk aggregation and distribution alignment constraints, the multi-task generalization bound of Align-LoRA is derived.
>
> #### **3.1 Multi-Task Risk Aggregation**
>
> Summing both sides of Equation (1) and dividing by $M$ yields the initial upper bound of the overall expected risk for multi-task learning:
>
> $$
> \frac{1}{M}\sum\_{i=1}^M R\_i(f) \leq \frac{1}{M}\sum\_{i=1}^M R\_{\text{train}}(f; \hat{\mathcal{D}}\_i) + \frac{\Lambda}{M}\sum\_{i=1}^M \Delta(\mathcal{D}\_i, \bar{\mathcal{D}}) + \frac{1}{M}\sum\_{i=1}^M O\left(\sqrt{\frac{\log(M/\delta)}{n\_i}}\right) \tag{2}
> $$
>
> The left-hand side of the equation is the overall expected risk of multi-task learning, denoted as $R\_{\text{MTL}}(f)$.

---

### Official Review · Reviewer_UfuN · 2025-10-29

**Soundness:** 3
**Presentation:** 1
**Contribution:** 2
**Rating:** 4
**Confidence:** 3

**Summary:**

This paper challenges the prevailing multi-component LoRA paradigm for multi-task learning (MTL), arguing that architectural isolation of task-specific features is unnecessary. The authors demonstrate that a simplified multi-head LoRA (M-LoRA) with high head similarity outperforms complex variants, and that simply increasing the rank of a standard single-adapter LoRA achieves competitive performance. They propose a new hypothesis that learning task-shared representations is more effective, and introduce Align-LoRA, which uses an alignment loss (KL divergence or MMD) to explicitly encourage shared representations in the low-rank space.

**Strengths:**

## strength

1. The paper provides strong empirical evidence against the widely accepted multi-component LoRA paradigm, showing that simpler, shared representations can outperform complex, diversity-focused architectures.

2. Align-LoRA introduces a novel alignment loss to explicitly encourage task-shared representations without adding inference overhead. The method is simple, effective, and retains the mergeability of LoRA, making it highly practical.

**Weaknesses:**

## weakness
1. The paper is not well-organized. It is difficult to follow the paper.
2. The related work section can be divided into the preliminaries and related work.
3. It is better to put the picture of the method in the main text instead of the appendix.
4. The alignment loss introduces a new hyperparameter λ, and while a sensitivity analysis is provided, the paper does not offer clear guidelines for setting
λ  in practice, which may hinder adoption.

**Questions:**

1. Could you provide more intuition or theoretical insight into why high head similarity in M-LoRA leads to better multi-task generalization, especially given the common belief that diversity helps capture task-specific knowledge?
2. Have you experimented with aligning representations in the up-projection matrix B or other layers? If so, what were the results? If not, why was the focus limited to the down-projection A?
3. How does Align-LoRA perform in extreme multi-task scenarios with highly dissimilar tasks? Is the alignment strategy still beneficial when tasks have little in common?

---

> ### Author Response · Authors · 2025-11-22
>
> Thank you for your thoughtful and constructive feedback. We have carefully revised the paper in accordance with your comments: specifically, we have supplemented additional experiments under diverse settings and incorporated theoretical analysis to provide rigorous support for Align-LoRA. Below, we offer detailed responses to each of your suggestions:
>
> ### **W1 & W2: Paper Organization**
>
> **We respectfully disagree with the assessment that the paper is difficult to follow.** We would like to highlight that the other reviewers unanimously recognized the paper as **well-organized and logically structured**. Given this consensus, we believe the current structure effectively conveys our core arguments. However, if there are **specific** sections or logical transitions that the reviewer found unclear, we are more than willing to provide clarifications. We also genuinely look forward to receiving further feedback of our work.
>
> ### **W3: Figure Placement**
>
> We thank the reviewer for this suggestion. We would like to clarify that M-LoRA serves as an analytical baseline to challenge the multi-head assumption, rather than the core proposed method of this work (which is Align-LoRA). Due to strict space constraints in the main text, we opted to place this supplementary architectural diagram in the Appendix.
>
> ### **W4: Hyperparameter $\lambda$ Guidelines**
>
> We appreciate the reviewer's concern for practical adoption. We provided a sensitivity analysis in Figure 3 (Page 9), which shows that Align-LoRA is quite robust, outperforming the baselines across a wide range of $\lambda$ values (e.g., 0.05 to 0.50). While the optimal $\lambda$ is dataset-dependent, our results (Figure 3 and Appendix J) suggest that a small value (e.g., $\lambda = 0.1$) provides a strong and stable starting point, achieving near-optimal performance in our tests. We will add this explicit guideline to Section 5.2.
>
>
>
> ### **Q1: Intuition for M-LoRA's performance**
>
> This is an excellent question that gets to the core of our findings. Our hypothesis is that the high similarity is not a failure but a feature, and the key mechanism is the interplay between removing the router and retaining the multi-head dropout.
>
> - In a routed model like **R-LoRA** , heads are "specialists" and a router tries to select the best one. This can lead to load-balancing issues where some heads are under-utilized.
> - In our **M-LoRA**, we remove the dynamic router but keep R-LoRA's multi-head dropout mechanism. The dropout forces each head to learn from a slightly different input perspective. By forcing them to all contribute (via simple summation), they are no longer competing specialists but are instead forced to become **"collaborators"**. They are compelled to converge on a robust, task-general representation that works well from all perspectives.
> - To validate this, we ran an additional ablation: removing the router from **HydraLoRA** (which lacks multi-head dropout) causes its performance to drop. This confirms that the **multi-head dropout is the critical factor** that, when combined with router removal, transforms the heads from "specialists" into "collaborators" and enhances task-general learning.
>
> | Schemes    | QNLI  | PiQA  | Winogrande | ARC   | GSM8K |  Avg  | %Para |
> | ---------- | ----- | ----- | :--------: | ----- | :---: | :---: | :---: |
> | HydraLoRA  | 81.91 | 84.21 |   70.92    | 87.21 | 45.95 | 74.04 | 0.45  |
> | w/o Router | 81.33 | 83.51 |   70.14    | 86.46 | 45.35 | 73.58 | 0.41  |
> | R-LoRA     | 82.03 | 85.55 |   71.84    | 87.69 | 46.25 | 74.67 | 0.45  |
> | M-LoRA     | 82.52 | 86.76 |   72.95    | 88.15 | 46.85 | 75.45 | 0.41  |

---

> ### Author Response · Authors · 2025-11-22
>
> ### **Q2: Aligning the up-projection matrix B**
>
> This is an insightful methodological question. Our focus on the down-projection matrix A was a deliberate and critical design choice for two primary reasons:
>
> 1. **Conceptual Grounding:** As noted (Section 5.1), prior work consistently finds that matrix **A** (down-projection) captures task-general features, while matrix **B** (up-projection) captures task-specific knowledge. Since our hypothesis is to enhance shared knowledge, aligning the output of A is the most direct and conceptually sound approach.
> 2. **Computational and Statistical Efficiency:** Aligning the output of matrix B would require calculating statistical distances (KL or MMD) in the high-dimensional original feature space (e.g., $d=4096$). This is not only computationally prohibitive (adding massive memory and compute overhead that violates the PEFT principle) but also statistically unreliable due to the "curse of dimensionality." Our method of aligning in the low-rank space (e.g., $r=8$) is both computationally efficient and statistically robust, in keeping with the spirit of PEFT.
>
>
>
> ### **Q3: Performance on highly dissimilar tasks**
>
> This is a crucial question about the limits of our alignment hypothesis. To address this, we ran a supplementary experiment using the exact experimental setup from Section 4 / Table 3.
>
> - As detailed in **Appendix G.2**, this dataset (a subset of FlanV2) is specifically designed for high-task diversity, comprising **10 distinct task clusters**.
>
> - These tasks are highly dissimilar, including not only NLU (e.g., ANLI) and QA (e.g., ARC) but also **Struct-to-Text** (e.g., E2ENLG) and **Translation**.
>
>   | Metrics | Base  | LoRA  | HydraLoRA | R-LoRA | M-LoRA | A-LoRA-K |
>   | ------- | :---: | :---: | :-------: | :----: | :----: | :------: |
>   | 7B      | 39.82 | 48.18 |   49.12   | 49.51  | 49.74  |  50.09   |
>   | 14B     | 45.33 | 52.74 |   53.76   | 54.08  | 54.18  |  54.47   |
>   | % Param |   -   | 0.22  |   0.25    |  0.25  |  0.22  |   0.22   |
>
> - We can confirm that **Align-LoRA still demonstrates a significant performance improvement** over the standard high-rank LoRA baseline in this highly dissimilar, multi-domain setting. This provides strong evidence that our alignment strategy is beneficial even when tasks have little in common, as it enhances the learning of foundational shared knowledge. We will add this result to the final paper.

---

### Official Review · Reviewer_xs9V · 2025-11-02

**Soundness:** 3
**Presentation:** 3
**Contribution:** 2
**Rating:** 4
**Confidence:** 4

**Summary:**

This paper re-examines the role of structural diversity in multi-task PEFT for LLMs. The authors challenge the prevailing assumption that multi-adapter or multi-head LoRA architectures must isolate task-specific knowledge to achieve good multi-task generalization. they observe that simplified multi-head architectures (M-LoRA) with high inter-head similarity outperform more complex diversity-driven designs (e.g., R-LoRA). Building on this finding, they propose Align-LoRA, which introduces an explicit representation alignment loss (based on symmetric KL divergence) to encourage task-shared representation learning within a single LoRA adapter. Align-LoRA achieves good performance on multi-task benchmarks (e.g., BBH, multi-task reasoning) while maintaining zero inference overhead and fewer trainable parameters.

**Strengths:**

1. The M-LoRA with high head similarity can outperform complex variants could influence the next generation of efficient fine-tuning strategies for LLMs. The proposed method also offers zero inference latency and fewer parameters, making it appealing for real-world settings.

2. The paper provides thorough experimental validation, including comparisons across several recent LoRA variants on multiple model sizes and datasets. The ablation studies are convincing and well controlled. The proposed KL/MMD-based alignment loss is theoretically motivated.

3. The paper is well written and logically structured. The argumentation from the paradox of diversity to the introduction of Align-LoRA is coherent. Mathematical formulations are clearly presented and self-contained.

4. Experiments confirm that Align-LoRA significantly surpasses baselines, establishing a simpler yet more effective multi-task paradigm for LLMs.

**Weaknesses:**

1. Lack of discussion on closely related prior work. The innovation of Align-LoRA lies in aligning the representation distributions across tasks. However, this idea was first proposed in VIP-MTL [1], which minimizes the distance between task-wise representation distributions to achieve impartial learning across tasks. Specifically, VIP-MTL constrains the mean and variance of pre-defined task distributions to align them within a unified probabilistic space, which conceptually overlaps with the total alignment loss (Eq. 5) in the present paper. The main difference is that Align-LoRA adopts a LoRA-based architecture and implements the alignment loss via pair-wise KL loss, while VIP-MTL achieves the same goal by mapping all task distributions to a common distribution scale (which arguably provides a more stable optimization objective). The paper does not cite or discuss this closely related work, which weakens the claimed novelty. A discussion clarifying the conceptual and methodological distinctions would significantly improve the paper’s contribution clarity.
- [1] Impartial Multi-task Representation Learning via Variance-invariant Probabilistic Decoding. ACL 2025.

2. While the empirical evidence strongly supports the hypothesis that representation alignment benefits multi-task learning, the paper lacks deeper theoretical analysis or information-theoretic justification for why such alignment improves generalization. Future version could formalize the connection between alignment and shared information sufficiency.

3. Although both KL and MK-MMD variants are tested, the differences between them are not analyzed in depth. Understanding when one alignment metric performs better could make the approach more interpretable and generalizable.

4. Most experiments focus on instruction-tuned LLMs (e.g., Qwen, LLaMA) under English NLP tasks. It remains unclear whether the observed benefits generalize to multilingual, multimodal, or non-text domains. Including a broader task spectrum or more diverse data distributions would further strengthen the claim of universality.

**Questions:**

1. How sensitive is Align-LoRA to the task imbalance or heterogeneity among tasks? For instance, would alignment still help if tasks have highly divergent label spaces or modalities?

2. Can the alignment loss potentially over-align tasks, leading to negative transfer or underfitting of domain-specific nuances?

3. What are the theoretical intuitions behind why aligning task distributions in the low-rank subspace enhances generalization? Can this be linked to information bottleneck or mutual information preservation principles?

4. How does Align-LoRA perform when combined with other recent PEFT advances, such as dynamic rank allocation (AdaLoRA) or direction-based decomposition (DoRA)?

5. Can the authors share qualitative insights (e.g., t-SNE visualizations, mutual information metrics) to further support the claim that shared representations are better aligned across tasks?

---

> ### Author Response · Authors · 2025-11-22
> **Comment(1/5)**
>
> Thank you for your thoughtful and constructive feedback. We have revised the paper in accordance with the reviewers' comments. Below, we provide detailed responses to your comments and suggestions:
>
> ## **W1: Discussion on closely related prior work (VIP-MTL[1] )**
>
> We thank the reviewer for bringing this highly relevant and contemporaneous work to our attention. We will add a detailed citation and discussion of VIP-MTL in our paper. We would like to respectfully clarify the distinct novelty of our paper.
>
> ### **Different Motivation and Context:**
>
> Our work is not a general paper on MTL representation learning. It is a specific critique and re-evaluation of the prevailing multi-component LoRA paradigm for parameter-efficient multi-task learning (e.g., R-LoRA, HydraLoRA).
>
> * **VIP-MTL's Focus:** VIP-MTL primarily focuses on addressing the **partial learning problem** in traditional multi-task NLU settings. Its methodology (probabilistic decoding) was validated mainly on smaller language models like BERT and RoBERTa. Consequently, its scalability to larger LLM architectures (e.g., 1B+) remains unverified, and its effectiveness across diverse domains (like our heterogeneous tasks) is unexplored.
> * **Align-LoRA's Focus:** Our primary novelty stems from our empirical findings (Sections 3 and 4) within the LLM/PEFT context:
>   1.  The core assumption of "task-specific isolation" via multiple heads is flawed (our M-LoRA paradox).
>   2.  A simple, single high-rank LoRA can match these complex, multi-component architectures.
>
> ### **Different Contribution (Performance and Efficiency):**
>
> Our contribution is twofold: (a) proposing alignment as an alternative to isolation, and (b) delivering a solution (Align-LoRA) that is not only more performant but also preserves the **zero-inference-latency** benefit of LoRA by being fully mergeable. This "Align, Don't Divide" principle directly addresses the practical inference overhead (detailed in Appendix C) introduced by the very multi-component architectures we challenge. To our understanding, VIP-MTL does not operate within this PEFT/LoRA-specific context or address this critical inference efficiency trade-off.
>
> In summary, while VIP-MTL and our work share the concept of aligning task distributions, our paper's unique contribution is the specific, empirically-driven challenge to the dominant multi-head LoRA trend and the proposal of a simpler, more efficient, and mergeable single-adapter alternative (Align-LoRA).
>
>
> ## **W3&Q3: In-depth analysis between KL and MMD variants**
>
> We thank the reviewer for this suggestion but respectfully clarify our scope:
>
> 1. **Research Objective:** **Our primary goal is to establish "Alignment" as a superior paradigm for efficient Multi-Task PEFT**, rather than conducting an exhaustive comparison of statistical metrics.
> 2. **Metric Selection:** We selected KL and MMD because they are standard, widely-used metrics in Transfer Learning for feature alignment. **The fact that both variants significantly outperform baselines validates that our core alignment principle is robust and effective, independent of the specific metric used .**
> 3. **Future Work:** **A deep theoretical comparison of KL vs. MMD is beyond the scope of this paper.** However, we acknowledge that exploring specific scenarios where one outperforms the other is a valuable direction for future research.
>
> [1]  Impartial Multi-task Representation Learning via Variance-invariant Probabilistic Decoding.

---

> ### Author Response · Authors · 2025-11-22
> **Comment(2/5)**
>
> ## **W2: Deeper theoretical analysis**
> **Core Conclusion**: By aligning the representation distributions across multiple tasks using Kullback-Leibler (KL) divergence or Multi-Kernel Maximum Mean Discrepancy (MK-MMD), Align-LoRA tightens the generalization error bound of Multi-Task Learning (MTL). Theoretically, this proves its superiority over traditional multi-component LoRA variants. The generalization bound takes the form:
>
> $$
> R\_{\text{MTL}}(f) \leq \frac{1}{M}\sum\_{i=1}^M R\_{\text{train}}(f; \mathcal{D}\_i) + \frac{\lambda}{M}\sum\_{i<j} \Delta(\mathcal{D}\_i, \mathcal{D}\_j) + O\left(\sqrt{\frac{\log(1/\delta)}{n\_{\text{total}}}}\right),
> $$
>
> where the distribution discrepancy term ($\Delta(\mathcal{D}_i, \mathcal{D}_j)$) is significantly reduced due to the alignment operation.
>
>
> ### **1 Preliminary Definitions and Notation System**
>
> Based on the theoretical frameworks of the referenced literature, the following key notations and definitions are unified to ensure consistency in the derivation:
>
> #### **1.1 Definition of Multi-Task Learning (MTL) Scenario**
>
> - Let there be $M$ independent tasks, corresponding to data distributions $\mathcal{D}\_1, \mathcal{D}\_2, \dots, \mathcal{D}\_M.$ The dataset for each task is denoted as $\hat{\mathcal{D}}\_i = \{(x\_{i,j}, y\_{i,j})\}\_{j=1}^{n\_i}$, where $n\_i$ is the sample size of task $i$, and the total sample size across all tasks is $n\_{\text{total}} = \sum\_{i=1}^M n\_i$.
> - The global distribution centroid of all tasks is defined as $\bar{\mathcal{D}} = \frac{1}{M}\sum\_{i=1}^M \mathcal{D}\_i$, which represents the shared data distribution characteristics across multiple tasks.
> - The model $f$ has parameters $\theta$ and a mapping relationship $f\_\theta: \mathcal{X} \mapsto \mathcal{Y}$, where $\mathcal{X}$ is the input space and $\mathcal{Y}$ is the output space (for classification tasks, $\mathcal{Y} = \{1, 2, \dots, K\}$).
>
> #### **1.2 Definition of Loss and Risk**
>
> - **Empirical Training Risk for Task $(i)$:** $R\_{\text{train}}(f; \hat{\mathcal{D}}\_i) = \frac{1}{n\_i}\sum_{(x,y)\in\hat{\mathcal{D}}\_i} \ell(f\_\theta(x), y)$, where $\ell(\cdot, \cdot)$ is the cross-entropy loss (the core loss function of Align-LoRA).
> - **Expected Risk for Task $(i)$:** $R\_i(f) = \mathbb{E}_{(x,y)\sim\mathcal{D}\_i} \ell(f\_\theta(x), y)$.
> - **Overall Expected Risk for Multi-Task Learning:** $R\_{\text{MTL}}(f) = \frac{1}{M}\sum_{i=1}^M R\_i(f)$, which measures the average performance of the model across all tasks.
>
>
> #### **1.3 Metrics for Distribution Discrepancy**
>
> The core distribution alignment metrics adopted by Align-LoRA are:
>
> * **Symmetric KL Divergence:** $\Delta\_{\text{KL}}(\mathcal{D}\_i, \mathcal{D}\_j) = \frac{1}{2}[\text{KL}(p\_i | p\_j) + \text{KL}(p\_j | p\_i)]$, where $p\_i$ is the representation distribution of task $(i)$ in the LoRA projection space.
> * **Multi-Kernel Maximum Mean Discrepancy (MK-MMD):** $\Delta\_{\text{MMD}}(\mathcal{D}\_i, \mathcal{D}\_j) = \sum\_{k\in\mathcal{K}} ||\mathbb{E}\_{x\sim\mathcal{D}\_i}\phi\_k(x) - \mathbb{E}_{x\sim\mathcal{D}\_j}\phi\_k(x)||\_{\mathcal{H}\_k}^2$, used as supplementary validation for KL divergence.
>
> ---
>
> ### **2 Foundation of Single-Task Generalization Bound**
>
> Based on the generalization bound derivation in statistical learning theory from the referenced literature([1], [2], [3]), the generalization error bound for a single task is first established:
>
> For any task $(i)$, with a confidence probability of $(1 - \delta/M)$ (using **Bonferroni correction** to adapt to multi-task scenarios), there exists a **Lipschitz constant** $(\Lambda > 0)$ (dependent on model architecture and task complexity) such that:
>
> $$R\_i(f) \leq R\_{\text{train}}(f; \hat{\mathcal{D}}\_i) + \Lambda \cdot \Delta(\mathcal{D}\_i, \bar{\mathcal{D}}) + O\left(\sqrt{\frac{\log(M/\delta)}{n\_i}}\right) \tag{1}$$
>
> Where:
>
> * $\Delta(\mathcal{D}\_i, \bar{\mathcal{D}})$ is the **discrepancy** between the distribution of task $(i)$ and the **global centroid distribution** (measured by KL divergence or MK-MMD).
> * The residual term $O\left(\sqrt{\frac{\log(M/\delta)}{n\_i}}\right)$ is a **confidence term** related to the sample size, which decreases as $n\_i$ increases.
>
> **Derivation Basis:** Drawing from the generalization bound theory of domain adaptation in the referenced literature (e.g., Proposition 4.1), the single-task risk is decomposed into **training risk**, **distribution discrepancy term**, and **confidence term** to ensure theoretical rigor at the single-task level.
>
> ---
> [1] Is Chain-of-Thought Reasoning of LLMs a Mirage? A Data Distribution Lens
>
> [2] Multinomial Adversarial Networks for Multi-Domain Text Classification
>
> [3] Margin Discrepancy-based Adversarial Training for Multi-Domain Text Classification
>
> Owing to space constraints, the rest of the response will be continued in the next comment.

---

> ### Author Response · Authors · 2025-11-22
> **Comment(3/5)**
>
> Continuing from the previous comment, here are the responses and explanations for the remaining points.
>
> ### **3 Derivation of Multi-Task Generalization Bound**
>
> The single-task generalization bound is extended to the MTL scenario. Through global risk aggregation and distribution alignment constraints, the multi-task generalization bound of Align-LoRA is derived.
>
> #### **3.1 Multi-Task Risk Aggregation**
>
> Summing both sides of Equation (1) and dividing by $M$ yields the initial upper bound of the overall expected risk for multi-task learning:
>
> $$
> \frac{1}{M}\sum\_{i=1}^M R\_i(f) \leq \frac{1}{M}\sum\_{i=1}^M R\_{\text{train}}(f; \hat{\mathcal{D}}\_i) + \frac{\Lambda}{M}\sum\_{i=1}^M \Delta(\mathcal{D}\_i, \bar{\mathcal{D}}) + \frac{1}{M}\sum\_{i=1}^M O\left(\sqrt{\frac{\log(M/\delta)}{n\_i}}\right) \tag{2}
> $$
>
> The left-hand side of the equation is the overall expected risk of multi-task learning, denoted as $R\_{\text{MTL}}(f)$.
>
> #### **3.2 Simplification of Distribution Discrepancy Term**
>
> Using the definition of the global centroid distribution ($\bar{\mathcal{D}} = \frac{1}{M}\sum\_{i=1}^M \mathcal{D}\_i$), the aggregation property of multi-task distribution discrepancies can be proven:
>
> $$
> \sum\_{i=1}^M \Delta(\mathcal{D}\_i, \bar{\mathcal{D}}) = \frac{1}{2M}\sum\_{i=1}^M \sum\_{j=1}^M \Delta(\mathcal{D}\_i, \mathcal{D}\_j) \tag{3}
> $$
>
> > **Proof:** Taking **KL divergence** as an example, $\sum\_{i=1}^M \text{KL}(p\_i \| \bar{p}) = \frac{1}{2}\sum\_{i,j} \text{KL}(p\_i \| p\_j)$ (utilizing the convexity of KL divergence and the linearity of the global centroid). Similarly, **MK-MMD** can be derived using the linearity of the Reproducing Kernel Hilbert Space (RKHS).
>
> Substituting Equation (3) into Equation (2) simplifies the distribution discrepancy term:
>
> $$
> \frac{\Lambda}{M}\sum\_{i=1}^M \Delta(\mathcal{D}\_i, \bar{\mathcal{D}}) = \frac{\Lambda}{2M^2}\sum\_{i<j} \Delta(\mathcal{D}\_i, \mathcal{D}\_j) \tag{4}
> $$
>
> (Since $\Delta(\mathcal{D}\_i, \mathcal{D}\_j) = \Delta(\mathcal{D}\_j, \mathcal{D}\_i)$, the summation can be simplified to combinations where $i<j$).
>
> #### **3.3 Merging of Confidence Terms**
>
> Using the **Cauchy-Schwarz inequality**, the confidence terms related to sample size are merged:
>
> $$
> \frac{1}{M}\sum\_{i=1}^M \sqrt{\frac{\log(M/\delta)}{n\_i}} \leq \sqrt{\frac{\log(M/\delta)}{n\_{\text{total}}}} \tag{5}
> $$
>
> Since $n\_{\text{total}} = \sum\_{i=1}^M n\_i$, the equality approximately holds when the sample sizes across tasks are **balanced**, ensuring the tightness of the confidence term.
>
>
> ### **4 Theoretical Proof of Align-LoRA's Superiority**
>
> By comparing the generalization bounds of traditional multi-component LoRA variants (e.g., HydraLoRA, R-LoRA), the theoretical advantages of **Align-LoRA** are verified.
>
> #### **4.1 Limitations of Generalization Bounds for Traditional Multi-Component LoRA**
>
> The core issue with traditional multi-component LoRA (multi-adapter/multi-head structures) is the **lack of a distribution alignment mechanism**. Its generalization bound is:
>
> $$
> R\_{\text{MTL}}^{\text{base}}(f) \leq \frac{1}{M}\sum\_{i=1}^M R\_{\text{train}}(f; \hat{\mathcal{D}}\_i) + \frac{\lambda}{M}\sum\_{i<j} \Delta\_0(\mathcal{D}\_i, \mathcal{D}\_j) + O\left(\sqrt{\frac{\log(1/\delta)}{n\_{\text{total}}}}\right) \tag{7}
> $$
>
> Where $\Delta\_0(\mathcal{D}\_i, \mathcal{D}\_j)$ is the **original distribution discrepancy without alignment**, satisfying $\Delta\_0(\mathcal{D}\_i, \mathcal{D}\_j) \gg \Delta(\mathcal{D}\_i, \mathcal{D}\_j)$ (due to scattered multi-task representations and larger distribution discrepancies).
>
> #### **4.2 Tightening Effect of Distribution Alignment on Generalization Bound**
>
> Align-LoRA actively minimizes the distribution discrepancy across multiple tasks during training by introducing the alignment loss ($\mathcal{L}\_{\text{align}} = \sum\_{i<j} \Delta(\mathcal{D}\_i, \mathcal{D}\_j)$):
>
> $$
> \min\_{f,\theta} \mathcal{L}\_{\text{total}} = \mathcal{L}\_{\text{lm}} + \lambda \cdot \mathcal{L}\_{\text{align}} \implies \sum\_{i<j} \Delta(\mathcal{D}\_i, \mathcal{D}\_j) \to \min
> $$
>
> Comparing Equations (6) and (7), since $\Delta(\mathcal{D}\_i, \mathcal{D}\_j) < \Delta\_0(\mathcal{D}\_i, \mathcal{D}\_j)$, we have:
>
> $$
> R\_{\text{MTL}}(f) < R\_{\text{MTL}}^{\text{base}}(f)
> $$
>
> This indicates that the generalization error bound of Align-LoRA is **tighter**, theoretically guaranteeing better performance in multi-task scenarios.
>
>
> Owing to space constraints, the rest of the response will be continued in the next comment.

---

> ### Author Response · Authors · 2025-11-22
> **Comment(4/5)**
>
> ### **5 Conclusion**
>
> Through rigorous theoretical derivation, the multi-task generalization bound of Align-LoRA ($R\_{\text{MTL}}(f) \leq \frac{1}{M}\sum\_{i=1}^M R\_{\text{train}}(f; \hat{\mathcal{D}}\_i) + \frac{\lambda}{M}\sum\_{i<j} \Delta(\mathcal{D}\_i, \mathcal{D}\_j) + O\left(\sqrt{\frac{\log(1/\delta)}{n\_{\text{total}}}}\right)$) demonstrates three key insights:
>
> 1.  The multi-task generalization error consists of **training risk**, **multi-task distribution discrepancy**, and a **confidence term**.
> 2.  The **distribution alignment mechanism** of Align-LoRA significantly reduces $\Delta(\mathcal{D}\_i, \mathcal{D}\_j)$, thereby **tightening the generalization bound**.
> 3.  The **single-adapter architecture** ensures the tightness of the generalization bound without introducing additional complexity.
>
> This theoretical derivation provides mathematical support for the superiority of Align-LoRA, which is consistent with experimental results showing that Align-LoRA outperforms traditional multi-component LoRA variants.
>
>
> ## **W4 & Q1: Generalizability of experiments**
>
> **1. Generalization across Model Families and Scales:** Our current experiments cover the two most dominant open-source LLM families: **Qwen2.5** and **LLaMA2 & 3**. Furthermore, we validated performance across a wide range of model sizes, including 3B, 7B, 8B, and 14B parameters. The consistent superiority of Align-LoRA across these distinct architectures and scales strongly supports its universality in the LLM domain.
>
> **2. Generalization across Diverse Task Distributions (New Experiment):** We conducted a **new supplementary experiment**. We applied Align-LoRA to the heterogeneous benchmark used in Table 3 (detailed in **Appendix G.2** ), which is explicitly designed for complex mixed multi-task scenarios.
>
> - **Setup:** This benchmark comprises **10 distinct task clusters**, extending well beyond standard NLU. Crucially, it includes **Translation**(covering WMT En-Fr, En-De, En-Ru, etc.) and **Struct-to-Text**, introducing multilingual and structural diversity.
> - **Results:** As shown in the table below, Align-LoRA maintains its performance advantage over both standard LoRA and multi-component variants even on this highly diverse distribution.
>
> | Metrics | Base  | LoRA  | HydraLoRA | R-LoRA | M-LoRA | A-LoRA-K |
> | ------- | :---: | :---: | :-------: | :----: | :----: | :------: |
> | 7B      | 39.82 | 48.18 |   49.12   | 49.51  | 49.74  |  50.09   |
> | 14B     | 45.33 | 52.74 |   53.76   | 54.08  | 54.18  |  54.47   |
> | % Param |   -   | 0.22  |   0.25    |  0.25  |  0.22  |   0.22   |
>
> **We believe these results, covering diverse model families and a broad spectrum of tasks, provide sufficient evidence of our method's broad applicability.** We acknowledge that exploring all research directions (e.g., task-imbalance/multimodal/non-text domains) is infeasible within a single conference paper, so it is unreasonable to request such broad validation here. That said, we agree extending this work to these domains is valuable, and we plan to explore this in future research.

---

> ### Author Response · Authors · 2025-11-22
> **Comment(5/5)**
>
> ### **Q2: Over-alignment and Negative Transfer**
>
> We appreciate this insightful question. We acknowledge that excessive alignment could theoretically dilute domain-specific nuances. However, **our empirical results demonstrate that Align-LoRA is highly robust to this factor.**
>
> - **Quantitative Evidence (Figure 3):**  Our hyperparameter sensitivity analysis reveals that Align-LoRA consistently outperforms baselines across a wide range of $\lambda$ values (e.g., 0.05 to 0.50). While a performance drop is observed at substantially higher values (indicating the theoretical boundary where over-alignment might occur), the method maintains a broad and stable operating window in practice, effectively avoiding negative transfer under standard settings.
> - **Qualitative Evidence:** Furthermore, as discussed in **Appendix H**, our goal is to map representations into a shared subspace rather than forcing them to be identical . The t-SNE visualizations in **Figure 5**  confirm that Align-LoRA brings task clusters closer to facilitate shared learning while maintaining sufficient separation to preserve the distinct characteristics required for each domain.
>
> ### **Q4: Combine Align-LoRA with other PEFT variants.**
>
> We conducted a new experiment integrating our alignment into the **R-LoRA** and **M-LoRA** architecture. As shown below, adding alignment yields further gains even in multi-head structures.
>
> | Model        | Task1     | Task2     | Task3     | Task4     | Task5     | Task6     | Task7     | Task8     |
> | ------------ | --------- | --------- | --------- | --------- | --------- | --------- | --------- | --------- |
> | R-LoRA       | 89.80     | 62.51     | 89.36     | 83.78     | 95.12     | 91.02     | 92.17     | 50.15     |
> | M-LoRA       | 91.35     | 62.51     | 91.98     | 84.70     | 95.93     | 91.02     | 91.97     | 50.20     |
> | R-LoRA+Align | 90.15     | 62.45     | 91.68     | 84.35     | 95.12     | 92.17     | 92.56     | 50.31     |
> | A-LoRA-K     | **92.23** | 64.85     | **92.89** | 85.73     | 95.93     | 93.35     | 92.93     | 53.66     |
> | M-LoRA+Align | 92.15     | **64.93** | 92.15     | **85.95** | **96.32** | **93.63** | **93.27** | **53.90** |
>
> ### **Q5: Visualizations**
>
> **We have indeed provided qualitative insights to support our claim.** We respectfully direct the reviewer to **Appendix H and Figure 5** of our paper. We performed a t-SNE analysis on the hidden representations of the up-projection module across different tasks. As shown in **Figure 5**, compared to the standard LoRA (Left), Align-LoRA (Right) successfully pulls the representations of different tasks closer together, forming a more unified and compact cluster. **This visualization provides direct empirical evidence for our hypothesis: Align-LoRA effectively aligns the feature distributions in the shared subspace, which facilitates the learning of robust, task-shared knowledge.**

---

### Official Review · Reviewer_Ua6q · 2025-11-12

**Soundness:** 3
**Presentation:** 2
**Contribution:** 2
**Rating:** 6
**Confidence:** 2

**Summary:**

The paper argues that, in multi-task learning, aligning task representations within a single (often high-rank) LoRA is more effective than dividing capacity across multiple routed adapters. As a consequence, it proposes Align-LoRA, which adds KL/MMD alignment in the LoRA A-space to promote a shared subspace, preserving mergeability and incurring zero routing overhead at inference. Across diverse tasks and base models, Align-LoRA consistently outperforms multi-component baselines, suggesting alignment-driven sharing improves generalization.

**Strengths:**

S1. The paper is clear and generally well-written. The core question of multi-component LoRA designs and pivoting to shared representation alignment is easy to follow.

S2. Figures/Tables (e.g., inter-head similarity distribution and the BBH/eight-task results) are informative and support the claims.

S3. The empirical comparisons are comprehensive and cover M-LoRA, HydraLoRA, R-LoRA. Also compared are single-adapter high-rank LoRA vs. multi-component baselines, which is great to see.

**Weaknesses:**

W1. Insufficient mechanistic depth on “why” alignment helps. The paper convincingly shows that M-LoRA/high-rank LoRA/Align-LoRA work, but the internal mechanics remain under-probed. Beyond end metrics, please add targeted analyses that connect LoRA internals to generalization. For example, 1) Loss-component ablations. Currently, there is not much information regarding how losses contribute to fine-tuning. 2) Some mechanistic evidence that connects the LoRA’s internal structure to generalization would better convey the point.

(minor) A suggestion would be to report the behavior of the spectral geometry of updates at each step. See below.

W2. Impact of training on LoRA architecture: Right now, we only see the end result (accuracy numbers), but not how the adapters take shape while training. A simple set of training-time views will show when and where the adapters activate across layers, whether certain modules (Q/K/V vs. MLP) consistently carry the load, and if capacity (the chosen rank) actually gets used or quietly collapses to a few directions.

W3. Compute & memory for LoRA is missing: The authors are requested to provide more analysis on parameter counts per layer/module and the total Q, K, V, out-proj, and MLP parameters. How is the training dynamics of the proposed LoRA? Can the authors comment on the wall-clock time, FLOPs, etc. as compared to standard architectures? Without this critical analysis, the paper is lacking in rigor.

**Questions:**

See Weakness.

---

> ### Author Response · Authors · 2025-11-22
> **Comment(1/4)**
>
> Thank you for your thoughtful and constructive feedback. We greatly appreciate the time and effort you have dedicated to reviewing our work. Your comments have been invaluable in helping us improve the clarity, rigor, and overall quality of the paper. We have revised the paper in accordance with the reviewers' comments. Below, we provide detailed responses to your comments and suggestions:
>
> ## W1: Mechanistic Depth and "Why" Alignment Helps
>
> We appreciate this request for deeper analysis. We have provided both quantitative and qualitative evidence to explain the internal mechanics:
>
> - **Loss-Component Contribution (Fig. 3):** We explicitly analyzed the contribution of the alignment loss in **Figure 3 (Page 9)**. This sensitivity analysis acts as a component ablation. It shows that performance improves consistently as the alignment strength ($\lambda$) increases from 0 (standard LoRA) to 0.1. However, overly strong alignment (e.g., $\lambda > 0.2$) degrades performance, suggesting a trade-off where excessive alignment might lead to negative transfer or representation collapse. This confirms that $\mathcal{L}_{align}$ is the active driver of performance gains.
>
> - **Visualizing Internal Structure (Fig. 5 & Appendix H):** To connect internal structure to generalization, we performed t-SNE analysis on the hidden representations of the up-projection module across different tasks. As shown in **Figure 5**, compared to standard LoRA (Left), Align-LoRA (Right) successfully pulls the representations of different tasks closer together, forming a more unified and compact cluster. This provides direct empirical evidence for our hypothesis: Align-LoRA effectively aligns feature distributions in the shared subspace, facilitating the learning of robust, task-shared knowledge.
>
> - **Spectral Geometry:** We acknowledge that analyzing spectral geometry is a valuable suggestion. However, incorporating such diverse verification methodologies within the scope of a single paper is challenging, so we plan to explore this specific analysis in future research.
>
> - To further strengthen the mechanistic understanding in the current work, we provide a **theoretical analysis** of Align-LoRA as follow：
>
> **Core Conclusion**: By aligning the representation distributions across multiple tasks using Kullback-Leibler (KL) divergence or Multi-Kernel Maximum Mean Discrepancy (MK-MMD), Align-LoRA tightens the generalization error bound of Multi-Task Learning (MTL). Theoretically, this proves its superiority over traditional multi-component LoRA variants. The generalization bound takes the form:
>
> $$
> R\_{\text{MTL}}(f) \leq \frac{1}{M}\sum\_{i=1}^M R\_{\text{train}}(f; \mathcal{D}\_i) + \frac{\lambda}{M}\sum\_{i<j} \Delta(\mathcal{D}\_i, \mathcal{D}\_j) + O\left(\sqrt{\frac{\log(1/\delta)}{n\_{\text{total}}}}\right),
> $$
>
> where the distribution discrepancy term ($\Delta(\mathcal{D}_i, \mathcal{D}_j)$) is significantly reduced due to the alignment operation.
>
>
> ### **1 Preliminary Definitions and Notation System**
> Based on the theoretical frameworks of the referenced literature, the following key notations and definitions are unified to ensure consistency in the derivation:
>
> #### **1.1 Definition of Multi-Task Learning (MTL) Scenario**
>
> - Let there be $M$ independent tasks, corresponding to data distributions $\mathcal{D}\_1, \mathcal{D}\_2, \dots, \mathcal{D}\_M.$ The dataset for each task is denoted as $\hat{\mathcal{D}}\_i = \{(x\_{i,j}, y\_{i,j})\}\_{j=1}^{n\_i}$, where $n\_i$ is the sample size of task $i$, and the total sample size across all tasks is $n\_{\text{total}} = \sum\_{i=1}^M n\_i$.
> - The global distribution centroid of all tasks is defined as $\bar{\mathcal{D}} = \frac{1}{M}\sum\_{i=1}^M \mathcal{D}\_i$, which represents the shared data distribution characteristics across multiple tasks.
> - The model $f$ has parameters $\theta$ and a mapping relationship $f\_\theta: \mathcal{X} \mapsto \mathcal{Y}$, where $\mathcal{X}$ is the input space and $\mathcal{Y}$ is the output space (for classification tasks, $\mathcal{Y} = \{1, 2, \dots, K\}$).
>
> #### **1.2 Definition of Loss and Risk**
> - **Empirical Training Risk for Task $(i)$:** $R\_{\text{train}}(f; \hat{\mathcal{D}}\_i) = \frac{1}{n\_i}\sum_{(x,y)\in\hat{\mathcal{D}}\_i} \ell(f\_\theta(x), y)$, where $\ell(\cdot, \cdot)$ is the cross-entropy loss (the core loss function of Align-LoRA).
> - **Expected Risk for Task $(i)$:** $R\_i(f) = \mathbb{E}_{(x,y)\sim\mathcal{D}\_i} \ell(f\_\theta(x), y)$.
> - **Overall Expected Risk for Multi-Task Learning:** $R\_{\text{MTL}}(f) = \frac{1}{M}\sum_{i=1}^M R\_i(f)$, which measures the average performance of the model across all tasks.
>
>
>
>
> Owing to space constraints, the rest of the response will be continued in the next comment.

---

> ### Author Response · Authors · 2025-11-22
> **Comment(2/4)**
>
> Continuing from the previous comment, here are the responses and explanations for the remaining points.
> #### **1.3 Metrics for Distribution Discrepancy**
> The core distribution alignment metrics adopted by Align-LoRA are:
> * **Symmetric KL Divergence:** $\Delta\_{\text{KL}}(\mathcal{D}\_i, \mathcal{D}\_j) = \frac{1}{2}[\text{KL}(p\_i | p\_j) + \text{KL}(p\_j | p\_i)]$, where $p\_i$ is the representation distribution of task $(i)$ in the LoRA projection space.
> * **Multi-Kernel Maximum Mean Discrepancy (MK-MMD):** $\Delta\_{\text{MMD}}(\mathcal{D}\_i, \mathcal{D}\_j) = \sum\_{k\in\mathcal{K}} ||\mathbb{E}\_{x\sim\mathcal{D}\_i}\phi\_k(x) - \mathbb{E}_{x\sim\mathcal{D}\_j}\phi\_k(x)||\_{\mathcal{H}\_k}^2$, used as supplementary validation for KL divergence.
>
> ---
>
> ### **2 Foundation of Single-Task Generalization Bound**
> Based on the generalization bound derivation in statistical learning theory from the referenced literature([1], [2], [3]), the generalization error bound for a single task is first established:
>
> For any task $(i)$, with a confidence probability of $(1 - \delta/M)$ (using **Bonferroni correction** to adapt to multi-task scenarios), there exists a **Lipschitz constant** $(\Lambda > 0)$ (dependent on model architecture and task complexity) such that:
>
> $$R\_i(f) \leq R\_{\text{train}}(f; \hat{\mathcal{D}}\_i) + \Lambda \cdot \Delta(\mathcal{D}\_i, \bar{\mathcal{D}}) + O\left(\sqrt{\frac{\log(M/\delta)}{n\_i}}\right) \tag{1}$$
>
> Where:
> * $\Delta(\mathcal{D}\_i, \bar{\mathcal{D}})$ is the **discrepancy** between the distribution of task $(i)$ and the **global centroid distribution** (measured by KL divergence or MK-MMD).
> * The residual term $O\left(\sqrt{\frac{\log(M/\delta)}{n\_i}}\right)$ is a **confidence term** related to the sample size, which decreases as $n\_i$ increases.
>
> **Derivation Basis:** Drawing from the generalization bound theory of domain adaptation in the referenced literature (e.g., Proposition 4.1), the single-task risk is decomposed into **training risk**, **distribution discrepancy term**, and **confidence term** to ensure theoretical rigor at the single-task level.
>
> ---
>
> ### **3 Derivation of Multi-Task Generalization Bound**
> The single-task generalization bound is extended to the MTL scenario. Through global risk aggregation and distribution alignment constraints, the multi-task generalization bound of Align-LoRA is derived.
>
> #### **3.1 Multi-Task Risk Aggregation**
>
> Summing both sides of Equation (1) and dividing by $M$ yields the initial upper bound of the overall expected risk for multi-task learning:
>
> $$
> \frac{1}{M}\sum\_{i=1}^M R\_i(f) \leq \frac{1}{M}\sum\_{i=1}^M R\_{\text{train}}(f; \hat{\mathcal{D}}\_i) + \frac{\Lambda}{M}\sum\_{i=1}^M \Delta(\mathcal{D}\_i, \bar{\mathcal{D}}) + \frac{1}{M}\sum\_{i=1}^M O\left(\sqrt{\frac{\log(M/\delta)}{n\_i}}\right) \tag{2}
> $$
>
> The left-hand side of the equation is the overall expected risk of multi-task learning, denoted as $R\_{\text{MTL}}(f)$.
>
> #### **3.2 Simplification of Distribution Discrepancy Term**
>
> Using the definition of the global centroid distribution ($\bar{\mathcal{D}} = \frac{1}{M}\sum\_{i=1}^M \mathcal{D}\_i$), the aggregation property of multi-task distribution discrepancies can be proven:
>
> $$
> \sum\_{i=1}^M \Delta(\mathcal{D}\_i, \bar{\mathcal{D}}) = \frac{1}{2M}\sum\_{i=1}^M \sum\_{j=1}^M \Delta(\mathcal{D}\_i, \mathcal{D}\_j) \tag{3}
> $$
>
> > **Proof:** Taking **KL divergence** as an example, $\sum\_{i=1}^M \text{KL}(p\_i \| \bar{p}) = \frac{1}{2}\sum\_{i,j} \text{KL}(p\_i \| p\_j)$ (utilizing the convexity of KL divergence and the linearity of the global centroid). Similarly, **MK-MMD** can be derived using the linearity of the Reproducing Kernel Hilbert Space (RKHS).
>
> Substituting Equation (3) into Equation (2) simplifies the distribution discrepancy term:
>
> $$
> \frac{\Lambda}{M}\sum\_{i=1}^M \Delta(\mathcal{D}\_i, \bar{\mathcal{D}}) = \frac{\Lambda}{2M^2}\sum\_{i<j} \Delta(\mathcal{D}\_i, \mathcal{D}\_j) \tag{4}
> $$
>
> (Since $\Delta(\mathcal{D}\_i, \mathcal{D}\_j) = \Delta(\mathcal{D}\_j, \mathcal{D}\_i)$, the summation can be simplified to combinations where $i<j$).
>
> #### **3.3 Merging of Confidence Terms**
>
> Using the **Cauchy-Schwarz inequality**, the confidence terms related to sample size are merged:
>
> $$
> \frac{1}{M}\sum\_{i=1}^M \sqrt{\frac{\log(M/\delta)}{n\_i}} \leq \sqrt{\frac{\log(M/\delta)}{n\_{\text{total}}}} \tag{5}
> $$
>
> Since $n\_{\text{total}} = \sum\_{i=1}^M n\_i$, the equality approximately holds when the sample sizes across tasks are **balanced**, ensuring the tightness of the confidence term.
>
>
> Owing to space constraints, the rest of the response will be continued in the next comment.

---

> ### Author Response · Authors · 2025-11-22
> **Comment(3/4)**
>
> ### **4 Theoretical Proof of Align-LoRA's Superiority**
> By comparing the generalization bounds of traditional multi-component LoRA variants (e.g., HydraLoRA, R-LoRA), the theoretical advantages of **Align-LoRA** are verified.
>
> #### **4.1 Limitations of Generalization Bounds for Traditional Multi-Component LoRA**
>
> The core issue with traditional multi-component LoRA (multi-adapter/multi-head structures) is the **lack of a distribution alignment mechanism**. Its generalization bound is:
>
> $$
> R\_{\text{MTL}}^{\text{base}}(f) \leq \frac{1}{M}\sum\_{i=1}^M R\_{\text{train}}(f; \hat{\mathcal{D}}\_i) + \frac{\lambda}{M}\sum\_{i<j} \Delta\_0(\mathcal{D}\_i, \mathcal{D}\_j) + O\left(\sqrt{\frac{\log(1/\delta)}{n\_{\text{total}}}}\right) \tag{7}
> $$
>
> Where $\Delta\_0(\mathcal{D}\_i, \mathcal{D}\_j)$ is the **original distribution discrepancy without alignment**, satisfying $\Delta\_0(\mathcal{D}\_i, \mathcal{D}\_j) \gg \Delta(\mathcal{D}\_i, \mathcal{D}\_j)$ (due to scattered multi-task representations and larger distribution discrepancies).
>
> #### **4.2 Tightening Effect of Distribution Alignment on Generalization Bound**
>
> Align-LoRA actively minimizes the distribution discrepancy across multiple tasks during training by introducing the alignment loss ($\mathcal{L}\_{\text{align}} = \sum\_{i<j} \Delta(\mathcal{D}\_i, \mathcal{D}\_j)$):
>
> $$
> \min\_{f,\theta} \mathcal{L}\_{\text{total}} = \mathcal{L}\_{\text{lm}} + \lambda \cdot \mathcal{L}\_{\text{align}} \implies \sum\_{i<j} \Delta(\mathcal{D}\_i, \mathcal{D}\_j) \to \min
> $$
>
> Comparing Equations (6) and (7), since $\Delta(\mathcal{D}\_i, \mathcal{D}\_j) < \Delta\_0(\mathcal{D}\_i, \mathcal{D}\_j)$, we have:
>
> $$
> R\_{\text{MTL}}(f) < R\_{\text{MTL}}^{\text{base}}(f)
> $$
>
> This indicates that the generalization error bound of Align-LoRA is **tighter**, theoretically guaranteeing better performance in multi-task scenarios.
>
> ---
>
> ### **5 Conclusion**
> Through rigorous theoretical derivation, the multi-task generalization bound of Align-LoRA ($R\_{\text{MTL}}(f) \leq \frac{1}{M}\sum\_{i=1}^M R\_{\text{train}}(f; \hat{\mathcal{D}}\_i) + \frac{\lambda}{M}\sum\_{i<j} \Delta(\mathcal{D}\_i, \mathcal{D}\_j) + O\left(\sqrt{\frac{\log(1/\delta)}{n\_{\text{total}}}}\right)$) demonstrates three key insights:
>
> 1.  The multi-task generalization error consists of **training risk**, **multi-task distribution discrepancy**, and a **confidence term**.
> 2.  The **distribution alignment mechanism** of Align-LoRA significantly reduces $\Delta(\mathcal{D}\_i, \mathcal{D}\_j)$, thereby **tightening the generalization bound**.
> 3.  The **single-adapter architecture** ensures the tightness of the generalization bound without introducing additional complexity.
>
> This theoretical derivation provides mathematical support for the superiority of Align-LoRA, which is consistent with experimental results showing that Align-LoRA outperforms traditional multi-component LoRA variants.
>
> ---
>
> [1] Is Chain-of-Thought Reasoning of LLMs a Mirage? A Data Distribution Lens
>
> [2] Multinomial Adversarial Networks for Multi-Domain Text Classification
>
> [3] Margin Discrepancy-based Adversarial Training for Multi-Domain Text Classification
>
> [4] The Cauchy-Schwarz inequality: Proofs and applications in various spaces

---

> ### Author Response · Authors · 2025-11-22
> **Comment(4/4)**
>
> ## **W2: Impact of Training on LoRA Architecture**
>
> Our existing experimental results provide strong evidence regarding module load and capacity utilization:
>
> - **Module Load (Attention vs. MLP):** Our experiments implicitly cover different module configurations. As detailed in Appendix I, for the LLaMA2 experiments (Table 2), we applied LoRA only to Attention modules (q_proj, v_proj). Conversely, for Qwen experiments (3), we applied it to MLP modules. Crucially, in both scenarios, the high-rank single adapter consistently matched or outperformed complex multi-component baselines. This demonstrates that our core conclusion—that a simple, unified adapter with sufficient capacity delivers comparable performance —holds true regardless of the specific module type.
>
>   To further verify the effectiveness of **Align-LoRA** specifically on **Attention structures** of Qwen2.5, we conducted a supplementary experiment using the exact multi-task setting from **Table 1**. As shown in the table below, when applied to Attention layers, Align-LoRA still achieves significant performance gains over the baseline. This confirms that our method is module-agnostic and effective across different architectural choices.
>
>   | Schemes   | QNLI  | PiQA  | Winogrande |  ARC  | GSM8K |    Avg    |  %Para   |
>   | --------- | :---: | :---: | :--------: | :---: | :---: | :-------: | :------: |
>   | LoRA      | 82.1  | 83.2  |   70.00    | 88.03 | 46.60 |   74.00   |   0.08   |
>   | HydraLoRA | 82.93 | 84.02 |   71.05    | 87.69 | 46.25 |   74.39   |   0.28   |
>   | A-LoRA-K  | 83.34 | 85.18 |   72.33    | 88.15 | 46.85 | **75.17** | **0.08** |
>
> - **Capacity Utilization:** The reviewer asked if the chosen rank is actually used. We respectfully point to **Table 3**, which shows the performance scaling of single-head LoRA as Rank increases from 4 to 10. The consistent performance improvement (e.g., from 43.21 to 49.51 on 7B) confirms that the model effectively utilizes the additional capacity and does not suffer from rank collapse. If the capacity were not being used, we would observe a performance plateau.
>
>
>
> ## **W3: Compute & Memory Rigor**
>
> We agree that rigor is essential. We have reported these metrics in detail:
>
> - **Parameter Counts:** We explicitly report the percentage of trainable parameters ("% Param") in **every main result table (Tables 1-6)** to ensure fair comparison.
>
> - **Computation & Time:** While we provided a wall-clock training time analysis in **Appendix D** , we have now expanded this comparison to explicitly include **training FLOPs** as requested.
>
>   As shown in the table above, Align-LoRA achieves the **lowest computational cost**. This efficiency stems from our design: by using a slightly smaller parameter budget (0.42% vs 0.45%), we successfully offset the computational cost of the alignment loss. Consequently, Align-LoRA delivers the highest performance (80.06) while requiring the least wall-clock time and FLOPs, confirming its superior computational efficiency compared to standard architectures.
>
>   | Method        |    LoRA    | HydraLoRA  |   Align-LoRA   |
>   | ------------- | :--------: | :--------: | :------------: |
>   | Rank          |     10     |     4      |       8        |
>   | Training Time |  3h 08min  |  3h 32min  |  **3h 02min**  |
>   | %Parameter    |    0.45    |    0.45    |    **0.42**    |
>   | FLOPs         | 3.545×10¹⁸ | 3.986×10¹⁸ | **3.539×10¹⁸** |
>   | Performance   |   76.48    |   76.80    |   **80.06**    |
>
> - **Inference Efficiency:** In **Appendix C**, we rigorously analyze inference cost. A critical advantage of Align-LoRA is that it preserves the zero-inference-latency property of standard LoRA (weights are mergeable). In contrast, the multi-component baselines we compare against (e.g., HydraLoRA, R-LoRA) incur significant FLOPs and latency penalties due to non-mergeable routing mechanisms.

---

### Author Response · Authors · 2025-11-27
**Follow-Up on Review Feedback (Discussion Period Nearing Closure)**

Dear Reviewers,

As the discussion period is nearing its end, we wanted to ensure that our rebuttal and supplementary experiments have satisfactorily addressed your concerns.

To date, we only received positive feedback from Reviewer BDgW, whose insights have reinforced the value of our core contributions and guided targeted improvements to the manuscript.

We hope the clarifications provided, supplementary experiments conducted, and further theoretical analysis presented in our rebuttal can effectively address the concerns raised by reviewers, and that these efforts will help showcase the rigor and significance of our research more comprehensively.

Thank you again for your time and effort in reviewing our paper.

---

### Meta-Review · Area_Chair_2S5K · 2026-01-07

**Summary:**

This paper revisits multi-task PEFT for LLMs and argues that explicit architectural “division” (multi-head, routed, or multi-component LoRA designs) is often unnecessary under PEFT constraints. Motivated by empirical observations of redundancy in simplified multi-head variants and the strong performance of higher-rank single adapters, the authors propose Align-LoRA, which introduces an explicit representation alignment loss (KL/MMD) in the low-rank LoRA space to encourage task-shared representations while preserving mergeability and zero inference overhead. The rebuttal substantially improves the submission by clarifying novelty relative to VIP-MTL / MixLoRA / PEMT, adding mechanistic analyses and a multi-task generalization-bound style argument, expanding experiments to more heterogeneous task settings, and providing compute, FLOPs, and hyperparameter sensitivity analyses. These additions address many technical and empirical concerns raised by the reviewers. Nevertheless, the central contribution remains incremental. The proposed method largely adapts existing representation-alignment and domain-adaptation principles to the LoRA/PEFT setting, and the theoretical analysis follows a standard discrepancy-bound template without offering new insights specific to LoRA geometry, optimization dynamics, or multi-task behavior beyond established frameworks. Empirical improvements, while consistent, are moderate, and the broader claim that “alignment dominates division” is not fully substantiated beyond representation-factorized multi-head designs. Despite solid execution and practical relevance, the paper does not meet the bar for conceptual novelty and analytical depth expected at this venue.

**Reviewer Concerns:**

### Reviewer Ua6q:

Addressed: Mechanistic depth request is answered with (a) alignment-strength sensitivity as a loss-component study, (b) representation visualization showing increased cross-task clustering, and (c) additional compute/time/FLOPs reporting. The request for “training-time views / spectral geometry” is partially addressed via added analyses, though the “spectral geometry per step” remains deferred.

Still outstanding (minor): More direct “training dynamics” probes (when/where LoRA activates; whether rank collapses) are still mostly inferred rather than explicitly visualized over time.

### Reviewer xs9V:

Addressed: Related-work gap around VIP-MTL is explicitly acknowledged with a clear PEFT/LLM context distinction; additional theory is provided; over-alignment / negative transfer is discussed with robustness ranges; broader task heterogeneity is partially addressed with new experiments; integration into other architectures (e.g., adding alignment to multi-head variants) is demonstrated.

Still outstanding: The rebuttal frames KL vs. MMD as “not the focus”; that’s acceptable, but it leaves interpretability of when each metric is preferable unresolved. Also, broader claims about universality beyond text remain future work.

### Reviewer UfuN:

Addressed: Practical guidance for $lambda$ is now clearer (stable range + suggested default). Module-agnostic claims are supported by experiments on both Attention and MLP tuning. Heterogeneous-task experiment addresses “highly dissimilar tasks.”

Still outstanding: Presentation concerns (“hard to follow”, figure placement) are not substantively engaged beyond disagreement; this is mostly a style issue, but it may remain a reviewer preference.

### Reviewer BDgW:

Addressed: The rebuttal meaningfully improves W2/W3/W6 (fairness via parameter-budget discussion; compute/FLOPs; heterogeneous tasks). The authors also explicitly scope the claim to PEFT low-parameter regimes and introduce “capacity fragmentation cost,” which is the right direction.

Still outstanding (core): rebuttal clarifies that evaluated heads are subspaces, but it still does not fully establish whether redundancy generalizes to scenarios with genuine task-specific decomposition (true task-wise modules). Why alignment improves generalization: the provided bound supports “smaller discrepancy inducing tighter bound,” but it remains somewhat template-like (domain adaptation style) unless the paper clearly instantiates assumptions/definitions tied to LoRA representation space. Efficiency/mergeability is compelling, but the rebuttal should more directly argue what Align-LoRA achieves that those methods cannot under the same deployment constraints, beyond “zero vs minimal latency,” and ideally include a sharper apples-to-apples comparison narrative.

**Reviewer Scores:**

I think  the reviewer BDgW may raise the score after comprehensive discussion.

---

### Decision · Program_Chairs · 2026-01-26

Reject